# Grounded Reinforcement Learning:
# Learning to Win the Game under Human Commands

**Shusheng Xu**[1]**, Huaijie Wang**[1] **and Yi Wu**[1,2]
[1] IIIS, Tsinghua University, Beijing, China
[2] Shanghai Qi Zhi Institute, Shanghai, China
{xuss20, wanghuai19}@mails.tsinghua.edu.cn
jxwuyi@gmail.com

## Abstract

We consider the problem of building a reinforcement learning (RL) agent that can both accomplish non-trivial tasks, like winning a real-time strategy game, and *strictly* follow high-level language commands from humans, like "attack", even if a command is sub-optimal. We call this novel yet important problem, *Grounded Reinforcement Learning* (GRL). Compared with other language grounding tasks, GRL is particularly non-trivial and cannot be simply solved by pure RL or behavior cloning (BC). From the RL perspective, it is extremely challenging to derive a precise reward function for human preferences since the commands are abstract and the valid behaviors are highly complicated and multi-modal. From the BC perspective, it is impossible to obtain perfect demonstrations since human strategies in complex games are typically sub-optimal. We tackle GRL via a simple, tractable, and practical constrained RL objective and develop an iterative RL algorithm, REinforced demonstration Distillation (RED), to obtain a strong GRL policy. We evaluate the policies derived by RED, BC and pure RL methods on a simplified real-time strategy game, MiniRTS. Experiment results and human studies show that the RED policy is able to consistently follow human commands and, at the same time, achieve a higher win rate than the baselines. We release our code and present more examples at `https://sites.google.com/view/grounded-rl`.

## 1 Introduction

Building assistive agents that can help humans accomplish complex tasks remains a long-standing challenge in artificial intelligence. Natural language, as the most generic protocol for humans to exchange and share knowledge, becomes a general and important interface for human-AI interaction. Decades of research efforts have been continuously made to develop AI systems that can ground agent behaviors to natural language commands [69, 64, 46].

Recently, as deep reinforcement learning (RL) techniques have been widely used to develop interactive agents to solve a variety of challenging problems, it becomes a new trend to cast language grounding as an RL problem to train agents that can automatically ground language concepts to visual objects or behaviors in an interactive fashion [29, 13, 15]. In particular, an RL agent is typically presented with a language command describing the goal of the RL task and will receive a success reward when the desired goal state under the command is achieved. A well-trained RL agent can often exhibit strong generalization capabilities to novel commands. Despite the simplicity and effectiveness, such an RL formulation requires a language generator to repeatedly output random commands and an explicit reward function to determine whether the command is accomplished. Hence, most existing RL works focus on simple navigation problems with template-based *compositional* languages over objects and attributes, e.g., "go to the red box next to the blue wall" [56, 68, 34, 15].

36th Conference on Neural Information Processing Systems (NeurIPS 2022).

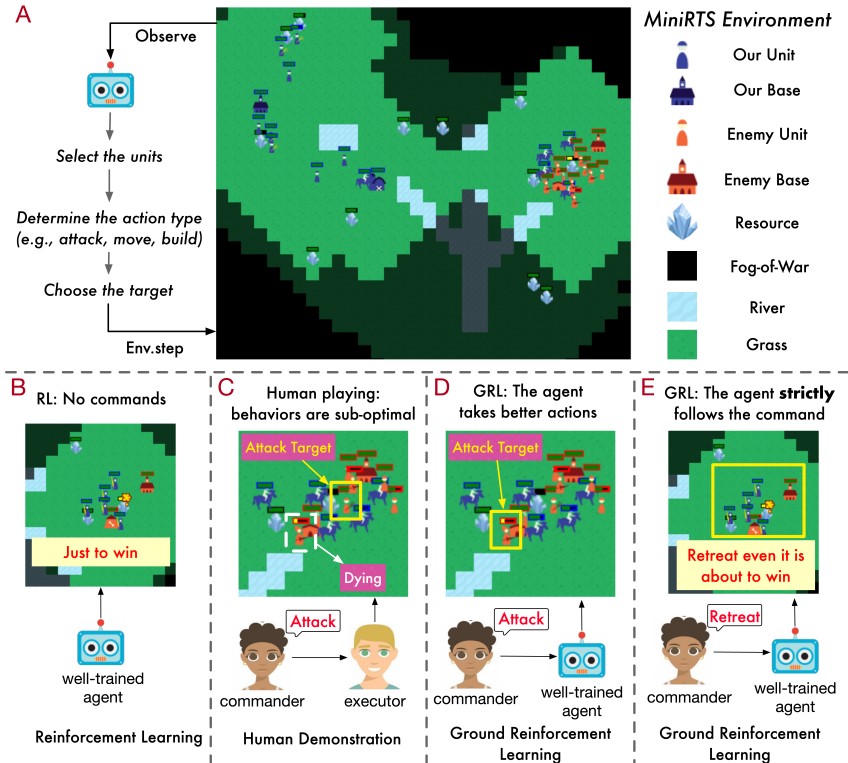

Figure 1: The grounded reinforcement learning (GRL) problem on the MiniRTS environment. *(A)* MiniRTS is a real-time strategy game where the player in blue needs to control its units to kill the enemy units in red. *(B)* A conventional RL agent. *(C)* MiniRTS provides a dataset of human demonstrations in the form of paired abstract language commands (e.g., "attack") and control action sequences. Human actions are often sub-optimal. *(D)* GRL aims to learn a command-conditioned agent such that it plays a winning strategy stronger than the human executor. *(E)* A GRL agent should *strictly* follow the human command even if it is sub-optimal.

To train agents that can interpret general human languages, another research direction is to leverage paired behavior-command data collected from human demonstrators. Representative applications include robotic manipulation, where the robot needs to map language commands to predefined motion skills [60, 35, 61, 65], and vision-and-language navigation, where an agent learns from human demonstrations to navigate towards a desired location in a 3D environment [51, 13, 29, 71]. Since an accurate command-following reward is no longer accessible, most works apply behavior cloning (BC) to directly imitate human behaviors or adopt inverse reinforcement learning (IRL) to learn a language-conditioned reward function [19, 52, 30, 23, 79, 17]. Although the use of human data enables natural commands, pure imitation-based methods require massive demonstrations, which can be expensive to collect. More importantly, both BC and IRL algorithms assume that the demonstrations are *optimal* [19, 52, 30, 23, 79, 17], in the sense that (i) each *behavior* demonstration is optimal under its paired language command, and (ii) the *language* commands are optimal for the underlying task. Such an assumption is reasonable in restricted domains like manipulation or navigation, where the optimal behavior is simply the shortest trajectory to the goal while human commands typically provide strong guidance towards task completion. However, in more complex problems like playing real-time strategy games, it is often the case that human players can be highly biased or sub-optimal [10].

Let's consider a concrete example in a simplified real-time strategy game, MiniRTS [33]. Fig. 1(C) shows a scenario where a human commander gives an abstract command "attack" to a human executor to attack enemy units. However, the action taken by the human executor is sub-optimal: a unit with full HP is under attack while a clear better strategy is to attack another dying unit. Moreover, the human commander can be sub-optimal as well. As shown in Fig. 1(E), very few enemy units are remaining, so the optimal command should be simply an "attack" for the win. However, the commander sends a

"retreat" command. This could be possibly due to the partially observed game state or human biases, but an obedient executor should still faithfully control the units to stop attacking and retreat.

We study this novel RL challenge, i.e., learning an strong agent capable of not only achieving a high win rate in a real-time strategy game, e.g., MiniRTS, but also *strictly* following high-level language commands, even if a *sub-optimal* command is given. We call this problem, *Grounded Reinforcement Learning* (GRL), which assumes an interactive RL environment with a dataset of (sub-optimal) human demonstrations reflecting proper human behaviors under language commands. In this setting, the rewards are not associated with the language commands, and it is non-trivial to verify whether a high-level language command is completed or not. So the policy can only learn about how humans follow commands from the demonstrations. We remark that GRL is different from standard language grounding problems since the primary mission is *not* towards better language understanding via interactions [11]. Instead, GRL focuses on the opposite direction, i.e., learning strong winning strategies while taking human commands as high-level behavior regulations, which is more related to the concept of developing human-compatible AI [57].

To tackle the GRL problem, we propose a simple yet effective constrained RL objective as a tractable approximation and developed an iterative RL algorithm, *REinforced demonstration Distillation* (RED), to derive a strong language-conditioned policy. RED adopts unconditioned RL training to encourage the policy to explore stronger winning strategies and periodically applies self-distillation and BC over demonstrations to ensure that the policy behavior is consistent with human preferences. We evaluate the performance of RED as well as baseline methods including BC-based methods and RL variants on the MiniRTS environment. Simulation results show that RED policy achieves strong command-following capabilities and a higher win rate than baselines under various types of test-time commands. We also conduct human evaluation by inviting 30 volunteers to play with policies trained by different methods. RED policy appears to be the most obedient under human votes and leads to a much higher collaborative win rate with human commanders.

## 2 Related Work

**Language grounding.** The idea of grounding languages to behaviors or concepts can be traced backed to 1970s [69], and most early works assume simple domain-specific languages and a pre-defined set of skills [45, 77, 37, 14, 47, 5]. Thanks to the recent advances of deep RL, it becomes feasible to ground languages to visual concepts and non-trivial behaviors in a simulated world. Many 3D environments with language-described goals have been developed over different domains, including maze exploration [15], visual navigation [13, 29, 71] and robot control [35, 61, 65], allowing end-to-end concept learning and grounding. RL training requires a goal generator and a reward function. Hence, these testbeds only adopt template-based languages over objects (e.g., box or cube) and attributes (e.g., spatial relation or color) and assume an oracle that can verify whether a state is consistent with the language description. We focus on following high-level natural commands.

Regarding natural commands, recent visual navigation benchmarks [3, 49, 61, 62] started to provide large-scale human demonstrations with the goal or pathway specified by natural languages. Behavior cloning (BC) is feasible since every human description is paired with a successful trajectory [19, 52, 30]. RL fine-tuning can be also applied, since a precise success measurement (i.e., distance to the goal position) w.r.t. each language instruction is known [22, 67]. Some works adopt inverse RL (IRL) to learn a language-conditioned reward function [23, 79] thanks to the fact that the demonstrations are actually optimal. We consider a more challenging game MiniRTS [33] with abstract language commands and a complex behavior space. BC and IRL methods work poorly in MiniRTS since both language commands and game trajectories from human annotators are highly sub-optimal.

There are also successful attempts to leverage powerful pretrained language models to map general languages to the goal space [36, 44] or a predefined set of skills [31, 60, 2] so that massive paired demonstrations can be no longer necessary. We primarily consider the problem of policy learning, so the use of pretrained model is orthogonal to our current focus but remains an exciting future direction.

There are also parallel works on language grounding in other domains, such as text games [38], answer questioning [4] and communication [39], where languages serve as state descriptor or action space rather than commands to follow. Our work is also related to ad-hoc team play [66, 12] since the policy cooperates with arbitrary commands. The difference is that the commander is not assumed to be optimal under the game reward. The policy must be assistive [28, 70] or even obedient [50].

**Constrained reinforcement learning.** We cast grounded RL as a constrained RL (CRL) formulation for tractability. CRL algorithms often leverage the Lagrangian multiplier [55] or projected gradients [1, 72] for policy improvement assuming simple (closed-form) constraints over states. In our setting, constraints can be only *implicitly* learned from human demonstrations. We simply adopt the behavior cloning objective, i.e., negative log likelihood, as the constraint satisfying metric, which is related to RL with demonstrations (RLwD) [32, 25]. However, due to limited data and a significant distribution shift between demonstrations and on-policy trajectories, RLwD methods can perform poorly. In addition, [73] iteratively optimizes the RL objective, projects the policy on a region around a reference policy, and enforces the policy satisfies the cost constraint. [27] updates the policies on the mixture of the expert replay buffer and the rollout trajectories. The most related work to our RED algorithm is *Supervised Seeded Iterated Learning* (SSIL) [42], which is designed for learning grounded communication and adopts a conceptually similar iterative framework by alternating between RL and self-distillation. SSIL suggests to jointly optimize RL and BC objectives assuming human communications are *perfect* demonstrations. By contrast, we find that an additional BC loss can be harmful to RL training due to the distribution shift issue. Similar empirical phenomena have been also reported in multi-task RL literature [75, 63, 74].

## 3 Preliminary

### 3.1 The MiniRTS Environment and Dataset

MiniRTS [33] is a grid-world RL environment (Fig. 1) that distills the key features of complex real-time strategy games. It has two parties, a player (blue) controlled by a human/policy against a built-in script AI (red). The player controls units to collect resources, do construction and kill all the enemy units or destroy the enemy base to win a game. A total of 7 unit types form a rock-paper-scissors attacking dynamics. Learning a strong policy is challenging due to partially observed states, complicated micro-management over dozens of units, and extremely diverse possible strategies.

In addition to an RL environment, MiniRTS provides a command-following dataset, which is collected by two humans playing collaboratively against the script opponent. One is the commander, who gives high-level strategic language commands like "attack", "retreat", or "build an archer". The other one is the executor, who controls the units to follow the language command. We remark that human data were collected against *much weaker* scripts than the default game opponent. Even in such an easier mode, the collected game episodes are highly sub-optimal with a win rate of merely 47.2%. The dataset contains a total of 63,285 commands with an average of 13.7 per game while the average game horizon is 151.7.[1] So, each single command may take a long horizon to accomplish. For notation simplicity in this paper, we assume an independent command is given to the executor at each time step, although a command should be repeated for a few steps. Full details are in appendix.

### 3.2 Notation

**Reinforcement learning.** We formulate the MiniRTS environment as a Partially Observable Markov Decision Process (POMDP), which is represented by a tuple $\langle \mathcal{S}, \mathcal{A}, \mathcal{O}, O, r, p \rangle$. $\mathcal{S}$ is the state space. $\mathcal{A}$ is the action space. $\mathcal{O}$ is the observation space. $r : \mathcal{S} \times \mathcal{A} \rightarrow \mathbb{R}$ is the reward function. $p : \mathcal{S} \times \mathcal{A} \times \mathcal{S} \rightarrow [0, 1]$ is the transition probability with $p(s'|s, a)$ denoting the probability from state $s$ to state $s'$ after taking action $a$. At each time step, the agent (i.e., the game player) observes $o_t = O(s_t)$, produces an action $a_t$ according to its policy $\pi_\theta$ parameterized by $\theta$, and receives a reward $r_t = r(s_t, a_t)$. The optimal policy should achieve high win rates in the game, namely $\theta^\star = \arg\max_\theta \mathbb{E}_{a_t, s_t} \left[ \sum_t r(s_t, a_t) \right]$. Note that we omit the discount factor $\gamma$ for conciseness only.

**Command-following policy.** We represent the policy by $\pi_\theta(a_t|o_t, c_t)$, which conditions on an observation $o_t \in \mathcal{O}$ and a language command $c_t \in \mathcal{C}$ at each time step $t$. $c_t$ can be generated from arbitrary distributions. $\mathcal{C}$ denotes the space of possible natural language commands. An LSTM is used to encode a command $c$ to an embedding. We use a special command "NA" to denote "*no command*", which corresponds to a zero embedding. When setting every $c_t$ to be NA, the command-following policy $\pi_\theta$ degenerates to a conventional policy in classical RL without language commands.

---

[1]Our numbers are slightly different from [33] because the original data-processing script is not released.

**Human demonstrations.** We denote human demonstrations as a dataset of trajectories, i.e., $\mathcal{D} = \{\tau_1, \tau_2, \ldots\}$. Each trajectory $\tau_i$, denoted by $\tau_i = (s_0^i, a_0^i, c_0^i, s_1^i, a_1^i, c_1^i, \ldots)$, consists of transitions from a single game played by a human executor and the paired command sequence from a human commander. Human behaviors can be sub-optimal: i.e., given the observation $o_t^i = O(s_t^i)$ and the command $c_t^i$, the chosen action $a_t^i$ can result in low rewards. Likewise, a command $c_t^i$ can be arbitrary w.r.t. human preferences. Finally, since human commanders are always asked to generate language commands, $\mathcal{D}$ does not contain any NA command, i.e., $c_t^i \neq \text{NA}$.

## 4 Method

### 4.1 Grounded Reinforcement Learning: Problem Formulation

The mission of GRL is to learn a policy $\pi_\theta$ such that $\pi_\theta$ can achieve high rewards in the environment and follow any possible command $c \in \mathcal{C}$ by achieving consistent behaviors with human demonstrations $\mathcal{D}$. We formulate the GRL problem as the following RL objective $J_G$:

$$J_G(\theta) = \mathbb{E}_{\substack{c_t \in \mathcal{C} \\ a_t \sim \pi_\theta(o_t, c_t)}} \left[ \sum_t r(s_t, a_t) \right] \quad \text{subject to} \quad K(\pi_\theta, \mathcal{D}) \leq \delta, \tag{1}$$

where $K$ denotes a distance metric between the policy behaviors from $\pi_\theta$ and the human data $\mathcal{D}$.

**Remark #1:** The commands can be arbitrary. $\pi_\theta$ needs to follow *any* possible command $c \in \mathcal{C}$ while still attains the highest rewards under the behavior constraint $K(\pi_\theta, \mathcal{D}) \leq \delta$.

**Remark #2:** Since the human actions in $\mathcal{D}$ can be sub-optimal w.r.t. the paired commands, it can be problematic to directly perform behavior cloning (BC) over $\mathcal{D}$. Therefore, we utilize a threshold $\delta$ so that the command-following policy $\pi_\theta$ can possibly deviate from the sub-optimal behaviors in $\mathcal{D}$.

Note that Eq. 1 is generally intractable and is presented only for the purpose of problem definition.

### 4.2 Constrained RL as a Tractable Approximation for GRL

There are two issues when directly optimizing Eq. (1). First, the command $c_t$ can be arbitrary. Second, the choice of behavior distance metric $K(\pi_\theta, \mathcal{D})$ should be specified.

**Training commands.** An obvious choice is to sample a random command $c_t$ from $\mathcal{C}$ at each time step $t$, which is perfectly aligned with Eq. (1). However, inconsistent commands from random sampling are typically harmful for a win, making the RL agent tend to ignore the commands. An alternative is a "human proxy" by learning a command model $h_\phi(c_t|s_t)$ over human data $\mathcal{D}$, leading to the objective $\mathbb{E}_{c_t \sim h_\phi(s_t), a_t \sim \pi_\theta(o_t, c_t)} \left[ \sum_t r(a_t, s_t) \right]$. Although this is reasonable, we need to be aware that human commands are sub-optimal and biased, which may further limit the chance for the policy $\pi_\theta$ to discover strong strategies during RL training. Thus, we propose an extreme version: i.e., directly performing unconditioned RL training by setting every command $c_t$ to be NA. This allows the policy to freely explore the strategy space during RL training for the best winning strategies.

**Command-following metric.** The simplest distance metric $K$ is the behavior cloning (BC) loss, i.e., negative log-likelihood (NLL) of $\pi_\theta$ for $(s_t, a_t, c_t) \in \mathcal{D}$. Specifically, we can define BC objective $L(\theta) = \mathbb{E}_\mathcal{D} \left[ -\log \pi_\theta(a_t|o_t, c_t) \right]$ and constrain $L(\theta) \leq \delta$ to ensure the policy behavior does not deviate from the demonstrations too much. There can be alternatives, such as KL-divergence or a learned metric. Regarding KL-divergence, since we are measuring the distance between samples $\mathcal{D}$ and a distribution $\pi_\theta$, $KL(\mathcal{D}||\pi_\theta)$ is equivalent to $L(\theta)$ while $KL(\pi_\theta||\mathcal{D})$ requires an behavior proxy over $\mathcal{D}$, which may involve additional biases. Regarding a learned metric, we can apply inverse RL to derive a reward function on whether the command is accomplished [23, 6, 7]. However, we empirically notice that a learning-based metric works poorly on MiniRTS. We hypothesis that this is because the demonstrations are sub-optimal and possibly limited in size for such a complex game.

**Tractable objective.** To sum up, we propose the following constrained RL objective $J_C(\theta)$ as a tractable objective for the GRL problem, i.e.,

$$J_C(\theta) = \mathbb{E}_{a_t \sim \pi_\theta(o_t, \text{NA})} \left[ \sum_t r(s_t, a_t) \right] \quad \text{subject to} \quad L(\theta; \mathcal{D}) = \mathbb{E}_\mathcal{D} \left[ -\log \pi_\theta(a_t|o_t, c_t) \right] \leq \delta. \tag{2}$$

Note that in Eq. (2), the sample distribution from the RL objective $J_C$ is *disjoint* from the BC objective $L(\theta; \mathcal{D})$ since we force $c_t = \text{NA}$ in RL training while $\mathcal{D}$ does not contain NA at all. Hence, Eq. (2) may look a bit ill-posed: an obvious solution is a "switching" policy which takes either the pure RL strategy or the BC strategy with or without a command. However, we will empirically show that such a switching policy does not provide strong performances. The insight is that we are training a conditioned *neural policy*. It is often observed in practice (e.g., multi-task learning) that a neural network can implicitly distill strategies from different modalities via its shared representations [20, 54, 76]. Therefore, the winning strategy discovered from RL training can empirically improve the sub-optimal command-following demonstrations within a bounded range. This phenomenon can be also justified by some recent theoretical findings [48, 16].

### 4.3 Reinforced Demonstration Distillation

A straightforward way to solve the constrained optimization problem in Eq. (2) is to adopt the Lagrangian multiplier, which leads to a joint optimization problem, i.e.,

$$J_C^{\text{soft}}(\theta) = J_C(\theta) - \beta L(\theta; \mathcal{D}). \tag{3}$$

Eq. (3) treats the BC objective $L(\theta; \mathcal{D})$ as an auxiliary loss of the stardard RL objective $J_C(\theta)$. The constraint in Eq. (2) can be satisfied by tuning the coefficient $\beta$. However, due to the distribution shift issue between on-policy samples and demonstrations, the supervised learning loss may interfere with the policy gradient and further makes the training process sensitive and unstable [21, 40].

**Iterative solution.** Inspired by the recent advances in self-imitation [53] and self-distillation [78, 43, 42], we adopt an iterative framework to decouple the RL loss and the BC loss in Eq. (3). The idea is simple: for RL training, we solely estimate the unconstrained policy gradient over $J_C$ without considering the BC loss; after policy improvement, we distill the demonstrations into the policy by running pure behavior cloning over both human demonstrations and self-generated winning trajectories. By repeatedly alternating between pure RL and pure BC, we are able to achieve a stable learning process. We call this method, REinforced demonstration Distillation (RED).

Specifically, let $\theta_k$ denote the parameters at iteration $k$, then we have the following update rule.

$$\text{RL phase:} \qquad \theta_k^{\text{RL}} \leftarrow \theta_{k-1} + \alpha \nabla J_C(\theta_{k-1}); \tag{4}$$

$$\text{BC phase:} \quad \theta_k \leftarrow \theta_k^{\text{RL}} - \alpha \nabla L(\theta_k^{\text{RL}}; \mathcal{D} \cup \mathcal{D}_k), \quad \mathcal{D}_k = \{\tau | \text{is\_win}(\tau), \tau \sim \pi_{\theta_k^{\text{RL}}}(\cdot, \text{NA})\}. \tag{5}$$

Here $\alpha$ denotes the learning rate. By repeating the update rules for $N$ iterations, we will derive our final parameter $\theta^\star$. The remaining issue is to ensure the constraint is satisfied, i.e., $L(\theta; \mathcal{D}) \leq \delta$. We empirically notice that this can be accomplished by controlling the ratio between the size of on-policy samples $\|\mathcal{D}_k\|$ and the size of demonstrations $\|\mathcal{D}\|$.

We present an intuitive justification of the self-distillation process from the perspective of sample distributions [9, 8]. We adopt RL for the highest rewards, so the unconditioned policy may frequently visit those states approaching a win. As a result, for games played by strong humans with aggressive commands, the paired game states will be more aligned with the RL state distribution while the states produced by biased or defensive human commanders will be more apart. As we are distilling two drift distributions into a single neural policy $\pi_\theta$, it will be more likely to get actions over those overlapping states improved within a bounded BC loss during optimization (see evidences in Sec. 5.3).

### 4.4 Implementations

We highlight some critical implementation factors here. More details are in appendix.

**Policy architecture.** We adopt a similar policy architecture to the provided executor model in MiniRTS [33] but with a modified action space and an additional value head as the critic. The original model outputs an action for *every* controllable unit, which makes RL training extremely slow. We introduce an additional 0/1 selection action on each unit denoting whether this unit *should act or not*, which substantially reduces the action dimension.

**Policy learning.** We use PPO [59] for RL training, which involves multiple mini-batch policy gradient steps. Since RL from scratch fails completely in practice, we fine-tune the BC policy and pretrain the value head with the policy parameters frozen. We also empirically find learning rate

warmup [41] particularly helpful for stabilizing training with a pretrained model: we start with learning rate 0 and then linearly increase it to the desired value $\alpha$.

**Creating $\mathcal{D}_k$.** Note that in the early stage of training phase, the winning rate can be low. In order to obtain sufficient data for $\mathcal{D}_k$, we implement a queue to store all past winning trajectories and use bootstrapped sampling if the queue is still not full. For the ratio between $\|\mathcal{D}_k\|$ and $\|\mathcal{D}\|$, we ensure the BC loss (i.e., negative log likelihood) of $\pi_\theta^\star$ over the validation set is at most 10% more than the pure BC policy (i.e., no RL training). We can also run sub-sampling over $\mathcal{D}$ to reduce the required sample size. Empirically, we find that simply setting $\|\mathcal{D}_k\| = \|\mathcal{D}\|$ leads to strong performances.

## 5 Experiment

All the simulation win rates are based on 1200 test games and repeated over 3 random seeds. More details and additional results including emergent behaviors are deferred to appendix.

**Baselines.** We consider the following methods in addition to our RED algorithm as baselines.
1. "*RL*" first pretrains the policy over demonstrations and then performs pure unconditioned RL training (i.e., conditioning on "NA" command) without any command-following constraint.
2. "*Switch*" is the "switching" policy, which consists of two policies, a pure BC policy from demonstrations and a pure RL policy without commands. When a langauge command is given, it runs the BC policy and uses the RL policy otherwise.
3. "*Joint*" optimizes the soft objective $J_C^{\text{soft}}$ in Eq. (3). $\beta$ is selected via a grid-search process.
4. "*IRL*" learns a reward function from demonstrations [24] and combines the game reward and the learned language reward to train a conditioned policy.

**Evaluation.** To evaluate a strong GRL policy, we need to answer the following questions.
1. *Does the policy learn a strong winning strategy?* The win rate can be a direct metric for this question. A RED policy should largely improves a pure BC policy with a high win rate.
2. *Does the policy follow the commands well?* A good GRL policy should keep its winning strategy while follow arbitrary commands. A precise evaluation of this question is non-trivial since the constraint satisfying condition, i.e., BC loss, can be a very misleading signal[2]. We adopt an approximate measurement via win rates in Sec. 5.1 and then conduct human evaluation in Sec. 5.4.

We also conduct study on OOD commands and algorithmic hyper-parameters in Sec. 5.2 and Sec. 5.3.

### 5.1 Main Results

**Q#1: is the RED policy strong?** We measure the win rates of different policies under two *test-time* command strategies, i.e., a pure `NA` strategy, which always gives an `NA`, and an *Oracle* strategy, which gives "optimal" commands. For the `NA` strategy, the GRL problem degenerates to the conventional RL setting, so a strong GRL policy should at least achieve comparable performances to the *RL* policy and substantially outperforms the BC (i.e., *Switch*) policy. For the *Oracle* strategy, it was scripted to give carefully tuned commands based on the ground-truth game state to instruct the policy to build dominating units according to the rock-paper-scissors dynamics. Hence, the GRL problem is converted to a fully cooperative game, and following *Oracle* commands becomes the optimal executor strategy. A strong GRL policy should achieve the highest possible win rate. Meanwhile, the policy with the highest win rate must produce a good command-following behavior. We also report the win rates of different methods with random commands and a "human proxy" commander, which is learned from human data. Since *Random* and *Human Proxy* are sub-optimal, a good GRL policy should have a substantially lower win rate than the `NA` case but still outperforms BC (*Switch*) policy.

The results are summarized in Tab. 1. The win rate by RED is the highest under oracle commands and is comparable to RL and Switch without commands (`NA`). Both RED and Joint policies substantially outperform Switch (BC) policy with sub-optimal commands (Random and Human Proxy), suggesting the conditioned behaviors are improved. Neither RL nor IRL is able to make the policy obedient – even under random commands, their win rates remain comparable to the `NA` case. We also note that commands from human proxy generally yield an even lower win rate than random commands. We remark that learning a good human commander model is extremely hard – not only due to the limited human command data in the dataset but also because most of the commands are highly sub-optimal.

---

[2]We empirically notice that a lower validation NLL does not necessarily yield a higher prediction accuracy.

|  | Command-Ignorant | | Command-Following | | |
| --- | --- | --- | --- | --- | --- |
| Test Commands | RL | IRL | Switch | Joint | RED |
| Oracle (%) | $89.3 \pm 0.7$ | $57.3 \pm 3.9$ | $78.7 \pm 0.9$ | $90.2 \pm 0.7$ | $\mathbf{92.6 \pm 0.6}$ |
| NA (%) | $57.5 \pm 1.1$ | $45.6 \pm 2.9$ | $57.5 \pm 1.1$ | $51.8 \pm 0.8$ | $\mathbf{57.8 \pm 1.3}$ |
| Random (%) | $53.3 \pm 0.4$ | $47.9 \pm 2.8$ | $12.3 \pm 0.2$ | $32.3 \pm 0.3$ | $29.8 \pm 0.8$ |
| Human Proxy (%) | $43.6 \pm 0.6$ | $48.9 \pm 2.9$ | $7.8 \pm 0.1$ | $11.4 \pm 0.7$ | $11.0 \pm 0.4$ |
| Adversarial Oracle (%) | $15.9 \pm 0.7$ | $35.0 \pm 2.4$ | $0.2 \pm 0.1$ | $0.4 \pm 0.2$ | $0.9 \pm 0.4$ |

Table 1: Win rates of different policies under various test-time commands. For Oracle and NA, a better policy should have a higher reward. For sub-optimal commands, i.e., Random and Human Proxy, an obedient policy should have a much lower win rate than the case of no commands (i.e., NA). For Adversarial Oracle, the win rates of command-following policies drop to almost 0.

**Q#2: is the RED policy grounded?**  Note that Tab. 1 has already provided evidences on the command-following performance of the RED policy, since the highest win rates can be only achieved by following the *Oracle* commands. We also implement an "*Adversarial Oracle*" commander, which always chooses the worst dominating units to build. We can observe that the win rates of Switch, Joint and RED drop to almost 0, which suggests that they follow commands even when the commands are divergent from the winning strategies.

Here we present an additional criterion: intuitively, as a command-following policy, its win rate should decrease if "worse" commands are fed; otherwise, if the win rate does not decay, the policy must not follow the commands well. Hence, we interpolate between the NA strategy and the full *Random* strategy and evaluate the win rate of different policies with different ratios of random commands. The results are shown in Fig. 2. We can observe that the RL policy and IRL policy are clearly not following the commands. Among the remaining command-following policies, RED policy generally produces a higher win rate than *Switch* and *Joint*, which provides evidences that our RED algorithm leads to improved strategies under the command-following requirement.

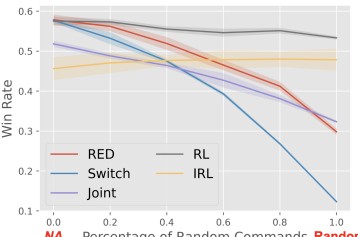

Figure 2: Win rates with an increasing amount of random commands. An obedient policy should have a decreasing win rate.

We also try to evaluate the command-following ability directly. At the beginning of a game, we provide a sequence of commands to guide the policies to build a specific army unit, and evaluate the success rates in 30 steps. The results are listed in Tab. 2. We can observe that the Switch, Joint, and RED policies achieve significantly higher success rates than IRL. This suggests that Switch, Joint, and RED faithfully follow the commands, while IRL tends to be command-ignorant.

|  | Command-Ignorant | | Command-Following | | |
| --- | --- | --- | --- | --- | --- |
| Unit Type | RL | IRL | Switch | Joint | RED |
| Spearman (%) | $98.9 \pm 0.2$ | $82.6 \pm 2.7$ | $90.6 \pm 0.9$ | $97.5 \pm 0.4$ | $96.0 \pm 1.4$ |
| Swordman (%) | $95.2 \pm 0.6$ | $70.4 \pm 5.2$ | $83.9 \pm 0.2$ | $96.2 \pm 0.7$ | $94.0 \pm 1.0$ |
| Cavalry (%) | $98.4 \pm 0.6$ | $80.4 \pm 7.1$ | $95.9 \pm 0.4$ | $98.9 \pm 0.2$ | $98.8 \pm 0.4$ |
| Dragon (%) | $81.8 \pm 1.6$ | $1.8 \pm 0.2$ | $83.6 \pm 0.8$ | $87.8 \pm 0.8$ | $88.8 \pm 2.2$ |
| Archer (%) | $85.7 \pm 0.6$ | $2.3 \pm 0.3$ | $96.3 \pm 0.1$ | $96.2 \pm 0.8$ | $94.6 \pm 1.5$ |
| Catapult (%) | $85.3 \pm 0.8$ | $2.2 \pm 0.9$ | $95.3 \pm 0.8$ | $96.2 \pm 0.2$ | $94.6 \pm 1.9$ |

Table 2: The success rates of building army units given the corresponding commands.

It is also worth noting that RL still achieves reasonable success rates. This is probably caused by the fact that RL policy is initialized with the BC policy and keeps a weak command-following ability. Building a single unit at the beginning of a game is relatively easy. However, following commands in a complex situation later in the game may be more difficult.

## 5.2   Out-of-Distribution Commands

As mentioned in Sec. 3.2, we adopt an LSTM encoder to encode arbitrary natural language commands into fixed-length sentence embeddings. During the training process, there are a total of 38,558

different commands in the training set. We also evaluate how the policies perform on Out-of-Distribution (OOD) commands. As shown in Tab. 3, we try 4 different ways of adding noise to *Oracle* commands and report the win rates. We can observe that the agents under noisy oracle commands perform better than *Random* commands but worse than *Oracle*. Intuitively, we can observe that the win rate roughly follows "*Random*" < "*Drop*" ∼ "*Replace*" < "*Shuffle*" ∼ "*Insert*" < "*Oracle*". This suggests that the LSTM encoder is sensitive to input words and word order. We also find that strong-following policies are affected more by noisy commands than weak-following policies, indicating that strong-following policies are more sensitive to the commands. In addition, we also investigate the influence of Out-of-Vocabulary (OOV) words in Appendix D.5, and visualize the command representations in Appendix D.6.

| | Command-Ignorant | | Command-Following | | |
|---|---|---|---|---|---|
| Noise Type | RL | IRL | Switch | Joint | RED |
| Random (%) | $53.3 \pm 0.4$ | $47.9 \pm 2.8$ | $12.3 \pm 0.2$ | $32.3 \pm 0.3$ | $29.8 \pm 0.8$ |
| Drop (%) | $61.9 \pm 1.0$ | $47.4 \pm 3.0$ | $38.8 \pm 0.6$ | $60.4 \pm 0.8$ | $61.1 \pm 0.6$ |
| Replace (%) | $59.1 \pm 2.4$ | $47.0 \pm 2.2$ | $27.6 \pm 0.9$ | $51.9 \pm 0.1$ | $56.7 \pm 1.8$ |
| Insert (%) | $75.4 \pm 0.7$ | $52.4 \pm 2.5$ | $78.3 \pm 0.7$ | $84.1 \pm 0.6$ | $81.1 \pm 3.1$ |
| Shuffle (%) | $75.1 \pm 1.8$ | $50.4 \pm 2.2$ | $68.6 \pm 1.4$ | $78.4 \pm 0.1$ | $77.9 \pm 0.5$ |
| Oracle (%) | $89.3 \pm 0.7$ | $57.3 \pm 3.9$ | $78.7 \pm 0.9$ | $90.2 \pm 0.7$ | $92.6 \pm 0.6$ |

Table 3: We add noise to **Oracle** commands. For Drop, we delete 50% of words in each command. For Replace, we replace 50% of the words with random words. For Insert, we insert random words and make the original command twice longer. For Shuffle, we shuffle the words in each command.

### 5.3 Ablation Study

**Training commands.** We test the performance of RED with different training commands in Fig. 3. The default RED yields the best strategy, i.e., the highest win rate with `NA` and oracle commands. Training with random commands or human proxy makes the policy command-ignorant – both two variants achieve similar win rates under `NA` and random commands.

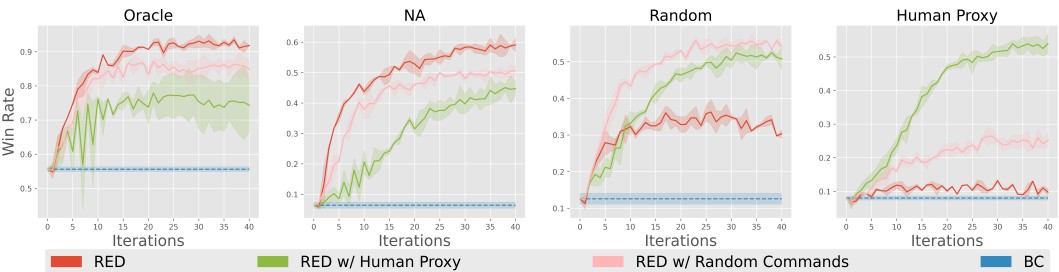

Figure 3: Win rates of RED with various training commands. RED has the strongest winning strategy with `NA` commands. Both random command and human proxy result in a command-ignoring policy.

**Ratio of $\|\mathcal{D}_k\|$ and $\|\mathcal{D}\|$.** We test different ratios of on-policy samples, i.e., $\|\mathcal{D}_k\|$, to the demonstration size, i.e., $\|\mathcal{D}\|$, in Tab. 4. We report the test-time win rates with oracle and `NA` commands as well as the constraint satisfaction condition, including both NLL (i.e., $L(\theta; \mathcal{D})$ from Eq. 2) and the action prediction accuracy, on the validation set. The results show that the validation NLL can be indeed controlled by tuning the dataset ratio, i.e., more RL samples consistently yielding a higher NLL. By contrast, we also find that a lower NLL does *NOT* necessarily lead to a higher action prediction

| $\|\mathcal{D}_k\| : \|\mathcal{D}\|$ | 2:1 | 1:1 | 1:0.5 | 1:0.25 |
|---|---|---|---|---|
| Test Win Rate with Different Commands | | | | |
| Oracle (%) | 87.7 | **92.6** | 92.0 | 89.6 |
| NA (%) | 54.6 | **57.8** | 56.1 | 57.0 |
| Constraint Satisfaction on Validation Set | | | | |
| NLL | 3.15 | 3.13 | 3.00 | 2.92 |
| Accuracy (%) | 68.9 | 68.5 | 69.0 | 69.2 |

Table 4: Ablation study on the ratio of $\|\mathcal{D}_k\|$ and $\|\mathcal{D}\|$. 0.5 means sub-sampling a half of data from $\mathcal{D}$. The data ratio leads to a controlled NLL.

accuracy. But we empirically find that simply setting a 1:1 ratio leads to a sufficiently good policy, which is the default choice for our RED algorithm.

**Learned strategy.** We identify the five most common categories of commands and measure the validation action prediction accuracy for both RED and BC policies to examine under what conditions the actions are changed – noting that RED policy uses BC policy as a warm-start. Fig. 4 shows the categorized accuracy differences with the dashed line denoting the overall mean difference. We can observe that RED policy disagrees with human actions the most under commands belonging to "attack" and "defend" categories, which both represent a highly complicated space of possible behaviors (e.g., which enemy unit to attack, how to defend). We believe these action changes are caused by policy improvement. While for commands with a low uncertainty belonging to "mine" and "stop" categories, RED policy simply follows humans. These findings provide empirical supports to our intuitive justification in Sec. 4.3.

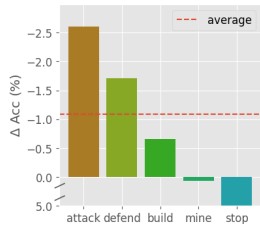

Figure 4: Action prediction accuracy difference between RED and BC for various command types.

## 5.4 Human Evaluation

We pick 4 policies, i.e., RED, Switch, Joint and RL, and invite 30 college student volunteers under department permission to complete a two-stage study. In the first stage, we shuffle the order of the 4 policies and ask each student to keep playing with the 4 policies using any commands. Once the student feels familiar with the game and policies enough, he/she is asked to rank the policies by the level of how they follow commands. In the second part, each student is asked to play 2 games per policy, and we measure the average winning rate of different policies.

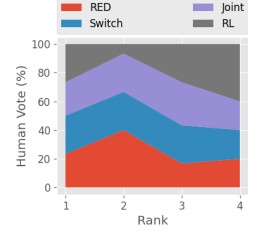

Figure 5: Human votes on command-following level. $x$: policy ranking. $y$: vote percentile.

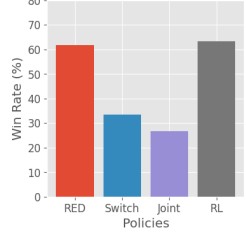

Figure 6: Human win rates when cooperating with 4 models. 60 games per model.

The results of ranking votes and win rates are shown in Fig. 5 and Fig. 6. Regarding the votes, the average rank of RED, Switch, Joint and RL are 2.3, 2.4, 2.5, 2.8, respectively. Fig. 5 visualizes the vote percentage for different policies. We notice a very interesting phenomenon that every policy has almost the same 1st-rank votes. Knowing that we shuffle the policy order while the students often spend a long time on the first policy to get familiar with MiniRTS, we hypothesize that the students are often biased towards the first policy they encountered. Nevertheless, RED has substantially more votes for the 2nd-rank, indicating a better command-following capacity. RL has the most 4th-rank votes, which is consistent with our expectation. Regarding the win rates (Fig. 6), RED outperforms Switch and Joint with a clear margin. The win rate of RL is comparable with RED. We believe this is due to the fact that RL policy often ignores human commands and simply acts for winning.

## 6 Conclusion

We tackle a new problem, grounded reinforcement learning (GRL), which aims to learn an agent that can not only get high rewards but also faithfully follow natural language commands from humans. We proposed a tractable objective, and developed an iterative RL algorithm RED, and evaluated the derived policy on a real-time strategy game MiniRTS. Both simulation and human results show that a RED policy achieves high win rates while exhibits strong command-following capabilities.

**Limitation and social impact.** As an initial study on the GRL problem, the RED algorithm is only evaluated on MiniRTS. This is because MiniRTS is the only public environment that is both complex (beyond "reaching goals") and provides natural language data. In addition, although we draw connections to existing theories in multi-task learning to justify why RED works, the theoretical principle of the underlying optimization process remains an open problem. Finally, even though it is in general debatable whether a robot should be fully obedient [18, 26, 50, 58], we believe that in restricted single-agent domains, like playing video games, strictly enforcing command-following behaviors should not result in worse negative social impact than BC or other RL methods.

### Acknowledgement

Yi Wu is supported by 2030 Innovation Megaprojects of China (Programme on New Generation Artificial Intelligence) Grant No. 2021AAA0150000.

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
