# Grounded Reinforcement Learning:
# Learning to Win the Game under Human Commands
# Supplementary Materials

**Project Website:** `https://sites.google.com/view/grounded-rl/`

## A MiniRTS Details & Dataset

In this section, we describe the details of MiniRTS Environment and human dataset. Our work is based on the game and dataset released at `https://github.com/facebookresearch/minirts` under CC BY-NC 4.0 license. The data do not contain any personally identifiable information or offensive content.

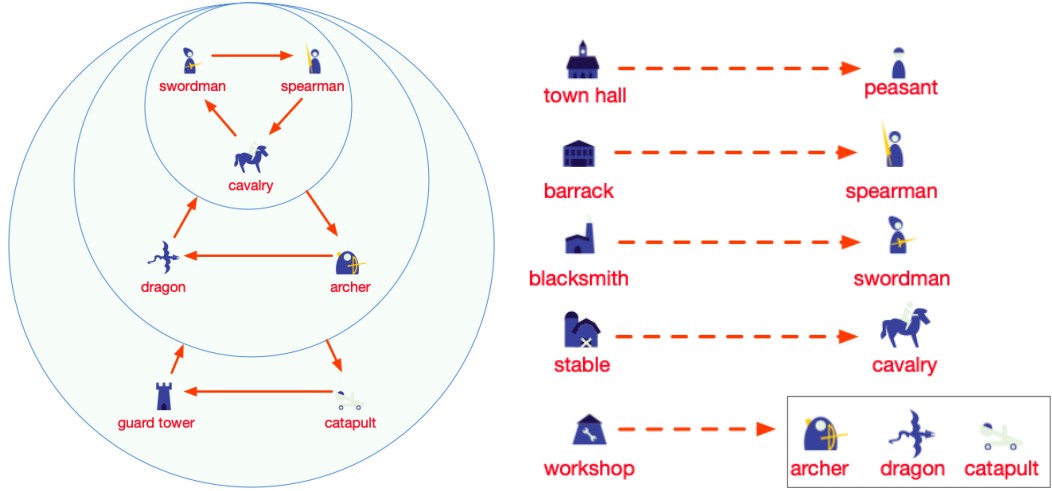

Figure 1: MiniRTS [2] implements the rock-paper-scissors attack graph, each army type has some units it is effective against and vulnerable to. For example, "swordman" restrains "spearman" but is retrained by "cavalry". "swordman", "spearman" and "cavalry" all are effective against "archer"

Figure 2: Building units can produce different army units using resources. "workshop" can produce "archer", "dragon" and "catapult" while other buildings can build one unit type. Only "peasant" can mine from resource units and construct building units.

### A.1 Game Design

**Game Units**   There are 3 kinds of units in MiniRTS, including resource units, building units, and army units.

- **Resource Units:** Resource units are stationary and neutral. Resource units cannot be constructed by anyone and are created at the beginning of a game. Only "peasant" (an

36th Conference on Neural Information Processing Systems (NeurIPS 2022).

| Action Type | Action Output |
|---|---|
| IDLE | NULL |
| CONTINUE | NULL |
| GATHER | ID of resource unit |
| ATTACK | ID of enemy unit |
| TRAIN | Army unit type |
| BUILD | Building unit type & (x, y) |
| MOVE | (x, y) |

Table 1: The action types and the corresponding action outputs.

army unit type) of both teams can mine from the resource units. One mine action could gather resources from the resource units, and the mined resources are necessary to build new building units or army units.

- **Building Units:** MiniRTS supports 6 different building unit types. 5 of the building unit types can produce particular army units by consuming resources (Fig. 2). The "guard tower" can not produce army units but can attack enemies. All the building units cannot move. Building units can be constructed by "peasant" at any available map location.

- **Army Units:** 7 types of army units can move and attack enemies. Specifically, a "peasant" can mine resources from resource units and construct building units with mined resources, but its attack power is low. The other 6 army unit types and "guard tower" are designed with a rock-paper-scissors dynamic. As shown in Fig. 1, each type has some units that it is effective against and vulnerable to.

**Game Map** The game map is a discrete grid of 32x32, where each cell can either be grass or river. The grass cell is available for constructing building units and is passable for any army unit, while the river cell only allows the "dragon" to go through. The map of each game is generated randomly. When initializing the map, one "town hall" and three "peasant" are placed for each player, then river cells and several resource units are added randomly. This generation phase ensures at least one path between two players' "town hall", and there are approximately equal resource units around each "town hall". In addition, the map is partially observable due to "fog-of-war".

## A.2 MiniRTS as an RL Environment

As an RL environment, a player is controlled by the RL agent while the opponent is a built-in script AI. The RL agent needs to control units to collect resources, do construction and kill all the enemy units or destroy the enemy base (i.e., "town hall") to win a game. We limit the length of a game to a maximum of 256 time steps.

**Observation Space** The observation of an agent includes a 32x32 map and extra states of the game (e.g., health points of the observable units, the amount of resources, etc.). The regions not visited are masked in the observation, and unseen enemy units are removed.

**Action Space** At each time step, the environment requires an action for each controlled unit, leading to a particularly large action space. We introduce an additional 0/1 action for each unit, denoting whether this unit should act or not. We generate the same action for those selected units, which reduces the action dimension substantially. In particular, our policy network outputs a *common* action as well as a 0/1 flag for *each* unit. Then we convert this into the standard MiniRTS format in the following way: for a unit assigned "1", it executes the common output action, for a unit assigned "0", it executes the action CONTINUE. After determining which units should act, following [2], we first predict an action type (e.g., MOVE, ATTACK), then predict the action outputs based on the action type. We summarize all available action types and their structure in Tab. 1. For IDLE, the selected units would do nothing. For CONTINUE, the units would continue their previous actions. For GATHER, we should also tell the units the ID of the target resource unit. For ATTACK, we should tell the units the enemy unit ID. For TRAIN, we should tell the units the army type they

should train. For BUILD, we should tell the units the building type and the position (x,y) to build. For MOVE, we should tell the units the target position.

**Reward**  This environment supports a sparse reward. At the end of a game, the reward is 1 if the agent wins and -1 if the agent loses. And the agent would receive the reward of 0 all the other time steps.

### A.3  Built-in Script AI

The authors of MiniRTS [2] provide several built-in script AIs. We find that an unconditioned RL agent achieves comparable win rates against both medium level and strong level script AIs, and we choose the medium level AI as the opponent for the convenience of designing the oracle commands. This script first sends all 3 initially available peasants to mine from the closest resource unit. It randomly chooses one army unit type from "spearman", "swordman", "cavalry", "archer", "dragon" and "catapult" and determines an army size $n$ between 3 and 7. It constructs a building unit corresponding to the selected army (Fig. 2), then trains $n$ units of the selected army type and sends them to attack. The script continuously trains the army units and maintains the army size of $n$.

### A.4  Dataset

The authors of MiniRTS [2] split the dataset into a training set and a validation set. The training set includes 4,171 trajectories, 57,293 commands and 634,799 transitions. While the validation set contains 433 trajectories, 5,992 commands, and 63,649 transitions. When training the policy, we only use data in the training set.

## B  Implementation Details

In this section, we introduce the implementation details and hyper-parameters. We train all the policies on a server with 8 RTX-3090 GPUs. All the policies warm-start with the BC pretrained model.

### B.1  RED Implementation

**Policy Architecture**  We adopt a similar policy architecture to the provided executor model in MiniRTS [2] but with a modified action space and an additional value head as the critic. The original model outputs an action for *every* controllable unit, which makes RL training extremely slow. We introduce an additional 0/1 selection action on each unit, denoting whether this unit *should act or not*, and only generate the same action for those selected units based on the average of their features, which substantially reduces the action dimension.

**RL Phase**  We train the policy through a parallel PPO training process. We use 1024 parallel workers to collect transitions $(s, a, r, s')$ from the environments synchronously ($c$ is NA for RED). Once 128 workers get 256 data points each, we split the collected data ($128 \times 256$ data points) into 4 batches and run one epoch of PPO training by setting the discount factor $\gamma$ as 0.999. We use GAE advantage for each data point. In each iteration, we repeat 100 training epochs as described above.

We collect winning trajectories and store them as $\mathcal{D}_k$. In the early stage of the training phase, the winning rate can be low. To obtain sufficient data for $\mathcal{D}_k$, we implement a queue to store all past winning trajectories and use bootstrapped sampling if the queue is still not full. Note that in each iteration, 100 epochs of PPO training produces $100 \times 128 \times 256 = 3,276,800$ transitions, and there are $634,799$ transitions in the training set of $\mathcal{D}$. We limit the capacity of $\mathcal{D}_k$ to the same as the size of $\mathcal{D}$. We report the results on different ratios of $\|\mathcal{D}_k\|$ and $\|\mathcal{D}\|$ in Sec. 5.2.

**BC Phase**  We adopt a similar BC training process to MiniRTS [2]. Since we introduce an additional 0/1 action to denote whether each controllable unit should act or not, we also need to train this action in the BC phase. In a transition of $\mathcal{D}$, each controllable unit is paired with an action type. We first find the most common action type across all units, then label the units with this action type as 1 and the rest units as 0. Finally, based on these labels, we train the additional 0/1 actions for all units. We run one epoch of BC training in each iteration by setting the batch size as $2048$.

**Optimizer & Learning Rate** We use Adam optimizer and adopt separate optimizers for RL and BC training. We run a total of 40 iterations (i.e. 4,000 PPO epochs.) to train a RED policy. For BC training, we set $\beta = (0.9, 0.999)$ and fix the learning rate as $2e - 4$. For RL training, we also set $\beta = (0.9, 0.999)$ but adapt the learning rate throughout all the 4,000 PPO epochs. In the first 500 epochs, we start with a learning rate of 0 and linearly increase it to the desired value $5e - 5$. In the rest 3500 epochs, we decrease the learning rate from $5e - 5$ to 0 linearly.

### B.2 Baseline Implementation

**RL Implementation** For pure RL baseline, we adopt the same architecture and hyper-parameters as RED. We run 4,000 PPO epochs. The batch size and number of transitions in each epoch are the same as RED. We also adopt the same optimizer and learning rate. The only difference is that there is no BC phase during the pure RL training. RL training also starts with the BC pretrained policy.

**Joint Implementation** For joint RL baseline, we modify the loss in the pure RL training by adding the NLL loss. The batch size to compute the NLL loss is the same as RL training. The architecture and all the hyper-parameters are the same as RED and pure RL training. We tune the weight $\beta$ of the NLL loss via a grid-search process, which is described in Sec. D.1.

**IRL Implementation** We adopt AIRL [1] for IRL training. The policy network has the same architecture and hyper-parameters as RED. We train a discriminator network $D_\phi(s, c, a)$ in the form of

$$D_\phi(s, c, a) = \frac{\exp f_\phi(s, c, a)}{\exp f_\phi(s, c, a) + \pi_\theta(a|s, c)}, \tag{1}$$

where $\pi_\theta$ is the policy network. The hyper-parameters are the same as the policy network. For the first 11 iterations, we run 25 discriminator epochs before each policy epoch. For the rest of the iterations, we run a single discriminator epoch before each policy epoch.

The architecture of $f_\phi(s, c, a)$ is similar to the policy network. We extract a fixed-length global feature vector, a fixed-length command feature vector, and fixed-length feature vectors for each unit and each position on the map in the same way as in MiniRTS [2]. Recall that an action contains an action type and possibly action outputs. We encode action type using an embedding layer. Action types IDLE and CONTINUE do not have any action output. For action types GATHER, ATTACK, and TRAIN, we encode action outputs by their corresponding unit embedding. For action type MOVE, we encode action outputs by the embedding of the moving location. For action type BUILD, we encode the building type using an embedding layer and then add it to the embedding of the building location to obtain the embedding of action output. The embedding of action is then obtained by concatenating the embedding of the action type and action outputs. In addition, we encode unit selection features by averaging the extracted features of the units that are selected. Finally, we concatenate the global feature, the command feature, the action feature, the unit selection feature, and the global continue flag together and feed it to a linear layer to get the value of $f_\phi(s, c, a)$.

The discriminator objective is given by

$$\mathbb{E}_\mathcal{D}[\log D_\phi(s, c, a)] + \mathbb{E}_{\phi_\theta}[\log(1 - D_\phi(s, c, a))]. \tag{2}$$

We combine environment rewards and the *intrinsic* rewards, namely

$$r(s, a) = r_{\text{env}}(s, a) + \beta_i \, \text{clip}\left(\frac{r_{\text{disc}}(s, c, a) - \mu}{\sigma}, -1, 1\right), \tag{3}$$

where

$$r_{\text{disc}}(s, c, a) = \log D_\phi(s, c, a) - \log(1 - D_\phi(s, c, a)), \tag{4}$$

and $\mu, \sigma$ are the mean and variance of $r_{\text{disc}}(s, c, a)$ within a sample batch, respectively. The tuning process of weight $\beta_i$ of the intrinsic reward is described in Sec. D.3.

## C  Experimental Details

### C.1  Oracle Commander

Since the built-in script AI only builds army units of a single type in a game. We script an oracle command strategy according to the ground truth of enemy units and the attack graph (Fig. 1). This

commander would ask the agent to build the appropriate army units step by step in the early stage of the game. Then, it sends `NA` and asks the policy to play on its own. A strong GRL policy would follow these commands to build the correct army units, play itself in the game's reset, and achieve a higher win rate.

- **Enemy "spearman":** (i) *"mine with all idle peasant"* in the first 2 time steps. (ii) *"build 3 peasant"* until there are 6 peasants. (iii) *"build a blacksimth"* until there is a blacksmith. (iv) *"build another swordman"* when the number of swordman is less than 5 otherwise `NA`.

- **Enemy "swordman":** (i) *"mine with all idle peasant"* in the first 2 time steps. (ii) *"build 3 peasant"* until there are 6 peasants. (iii) *"build a stable"* until there is a stable. (iv) *"build another cavalry"* when the number of cavalry is less than 5 otherwise `NA`.

- **Enemy "cavalry":** (i) *"mine with all idle peasant"* in the first 2 time steps. (ii) *"build 3 peasant"* until there are 6 peasants. (iii) *"build a barrack"* until there is a barrack. (iv) *"build a spearman"* when the number of swordman is less than 5 otherwise `NA`.

- **Enemy "dragon":** (i) *"mine with all idle peasant"* in the first 2 time steps. (ii) *"build 3 peasant"* until there are 6 peasants. (iii) *"build a workshop"* until there is a workshop. (iv) *"make archer"* when the number of archer is less than 5 otherwise `NA`.

- **Enemy "archer":** (i) *"mine with all idle peasant"* in the first 2 time steps. (ii) *"build 3 peasant"* until there are 6 peasants. (iii) Randomly select a building command from *"build a blacksimth", "build a stable"* and *"build a barrack"* until the corresponding building unit is constructed. (iv) Select from *"build another swordman", "build another cavalry"* and *"build a spearman"* according to the constructed building unit when the number of the corresponding army unit is less than 5 otherwise `NA`.

- **Enemy "catapult":** We empirically find that the best strategy to deal with the "catapult" is to send `NA` all the time (according to the policies of Joint, RED and Switch). We hypothesize it is because that all the other army types are effective against "catapult", and building commands as before are not helpful to play against catapult.

## C.2 "Human Proxy" Command Generator

Training a command generator is not our focus. We use the best commander (instructor) model trained and released by MiniRTS [2]. Please refer to the model at `https://github.com/facebookresearch/minirts`.

## C.3 Learned Strategy

**How to Categorize Commands**  We identify the five most common categories of commands as follows:

- **Attack**: Commands containing "attack" or "kill" belong to this category. This type of command typically instructs the agent to attack enemy units, but does not necessarily specify which unit to attack.

- **Defend**: Commands containing "defend" belong to this category. This type of command typically asks the agent to fight with invading enemy units but usually does not give more detail.

- **Build**: Commands containing "build", "create" or "make" belong to this category. This type of command typically instructs the model to build a specific type of building unit or army unit.

- **Mine**: Commands containing "mine", "mining" or "mineral" belong to this category. This type of command typically instructs the model to mine from the resource units.

- **Stop**: Commands containing "stop" belong to this category. Such commands typically claim that the current action should be terminated.

**Action Prediction Accuracy**  The actions in the transitions of $\mathcal{D}$ are extremely complex. In each transition, there are many controllable units, and each unit has its action type and the corresponding action output (see Tab. 1). So it is not so direct to compute the action prediction accuracy. When

predicting the action prediction, we concentrate on the units whose action types are not CONTINUE, since the units with CONTINUE would continue their previous actions, which should be considered in the previous transitions. In detail, for a single transition, we skip the actions of determining whether the units should act or not, and treat the units whose actions are *not* CONTINUE as the selected units. We predict the action type and action output based on the selected units. We think the prediction is correct if the predicted action type and action output exactly match with any selected unit. The accuracy is then computed as the correct prediction ratio over all the transitions. In addition to action prediction accuracy , we also evaluate NLL difference of RED policy and BC policy in Sec. D.4.

# D   Additional Results

## D.1   Grid Search on $\beta$ for Baseline "Joint"

We conduct the experiments on joint training with different $\beta$, and the results are listed in Tab. 2. We can observe that the policy with a larger $\beta$ performs worse when no commands are provided (NA), which indicates that the joint objective may be harmful to RL training. In addition, the win rates tested with random commands suggest that a smaller $\beta$ may result in the policy being less obedient.

| $\beta$ | 0 | 0.5 | 1 | 2 |
|---|---|---|---|---|
| Oracle (%) | $89.3 \pm 0.7$ | $89.9 \pm 0.3$ | $\mathbf{90.2 \pm 0.7}$ | $87.8 \pm 0.8$ |
| NA (%) | $\mathbf{57.5 \pm 1.1}$ | $54.9 \pm 0.9$ | $51.8 \pm 0.8$ | $48.4 \pm 0.6$ |
| Random (%) | $53.3 \pm 0.4$ | $34.3 \pm 0.9$ | $32.3 \pm 0.3$ | $28.9 \pm 0.2$ |
| Human Proxy (%) | $43.6 \pm 0.6$ | $12.1 \pm 0.7$ | $11.4 \pm 0.7$ | $10.4 \pm 0.4$ |

Table 2: Results of Joint on different $\beta$, where $\beta = 0$ is equal to pure RL training.

## D.2   Adaptive $\beta$

Although it is empirically a common practice to use a fixed $\beta$ in the Lagrangian method, we additionally provide experiments in which the coefficient $\beta$ is updated during the training process. It is worth noting that a similar technique can also be applied to our proposed method RED. In particular, during the BC phase, we add a coefficient $\beta$ on the behavior clone loss on demonstrations $\mathcal{D}$. This coefficient $\beta$ is updated during the training process. The results are listed in Tab. 3.

| | Switch | Joint (fixed $\beta$) | RED (fixed $\beta$) | Joint (adaptive $\beta$) | RED (adaptive $\beta$) |
|---|---|---|---|---|---|
| Oracle (%) | $78.7 \pm 0.9$ | $90.2 \pm 0.7$ | $92.6 \pm 0.6$ | $\mathbf{93.7 \pm 0.1}$ | $93.1 \pm 0.1$ |
| NA (%) | $57.5 \pm 1.1$ | $51.8 \pm 0.8$ | $57.8 \pm 1.3$ | $57.7 \pm 0.3$ | $\mathbf{61.2 \pm 0.3}$ |
| Random (%) | $12.3 \pm 0.2$ | $32.3 \pm 0.3$ | $29.8 \pm 0.8$ | $38.1 \pm 1.1$ | $37.6 \pm 1.2$ |

Table 3: Results of Joint and RED using adaptive $\beta$.

One can see that using an adaptive $\beta$ improves the performance of both Joint and RED. While having similar performance under the Oracle commander, RED with adaptive $\beta$ significantly outperforms Joint under the NA commander. This suggests that iterating between RL and BC ensures better RL performance than mixing the two objectives.

## D.3   Grid Search on $\beta_i$ for Baseline "IRL"

We adopt AIRL [1] as the algorithm for IRL experiments. We learn a reward function from demonstrations and combine the game reward and the learned language reward to train a conditioned policy (See Eq. 3). We use the human proxy command generator to provide commands when training, since the learned language reward is conditional on game states and human commands. We evaluate the performance with different $\beta_i$. and the results are listed in Tab. 4

| $\beta_i$ | 0.001 | 0.005 | 0.01 |
|---|---|---|---|
| Oracle (%) | **57.3 ± 3.9** | 53.7 ± 2.8 | 43.2 ± 2.0 |
| NA (%) | **45.6 ± 2.9** | 41.5 ± 2.8 | 33.7 ± 3.3 |
| Random (%) | 47.9 ± 2.8 | 43.9 ± 1.2 | 30.1 ± 2.1 |
| Human Proxy (%) | 48.0 ± 2.9 | 43.6 ± 3.4 | 33.9 ± 2.2 |

Table 4: Results of IRL on different $\beta_i$.

We can observe that the win rates tested with random commands are comparable with tested with NA, which indicates that IRL is not able to make the policy obedient, although we increase the weight of the learned reward. In addition, the learned reward may be harmful to the RL training, larger $\beta_i$ leads to lower win rates tested on the oracle and NA commands.

### D.4 Validation NLL for Various Command Types

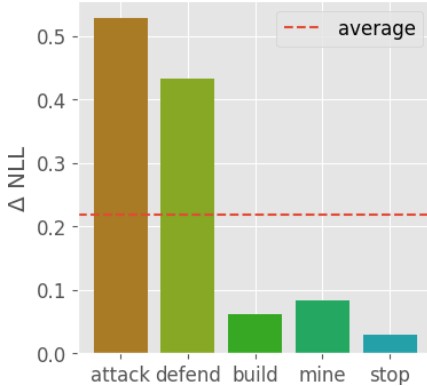

Figure 3: Validation NLL between RED and BC for various command types.

In addition to action prediction accuracy, we also evaluate NLL difference of RED policy and BC policy in Fig. 3, on the validation set. The dashed line denotes the overall validation difference. The conclusion is the same as action prediction accuracy. RED disagrees with the commands with a highly complicated space of possible behaviors (e.g., the command belonging "attack", "defend") the most, and follows humans' action on the commands with low uncertainty (e.g., the commands belonging to "stop").

### D.5 Performance on Commands with OOV Words

We also evaluate the policy performances on commands with Out-of-Vocabulary (OOV) words. It is worth noting that we use a special token [unk] to handle all the OOV words. We insert the special token in different positions of the *oracle* commands to evaluate the influence of OOV words. We describe the ways to insert the OOV words as follows. And the results are listed in Tab. 5.

- All-OOV: We replace all words with [unk] tokens in each command.
- Pre-OOV: We insert a sequence of [unk] tokens in front of each command, which makes the command twice the length of the original.
- Post-OOV: We insert a sequence of [unk] tokens at the end of each command, which makes the command twice the length of the original.
- Mid-OOV: We insert [unk] tokens between any two words of each command.

The results show that "All-OOV" commands are terrible for our strong following policies (i.e., RED, Joint, and Switch). RL and IRL policies perform near the same under the *NA* command and *All-OOV* commands, which suggests they are command-ignorant. Performances on *Pre-OOV*, *Post-OOV* and

| Test Commands | Command-Ignorant | | Command-Following | | |
|---|---|---|---|---|---|
| | RL | IRL | Switch | Joint | RED |
| Oracle (%) | $89.3 \pm 0.7$ | $57.3 \pm 3.9$ | $78.7 \pm 0.9$ | $90.2 \pm 0.7$ | $92.6 \pm 0.6$ |
| NA (%) | $57.5 \pm 1.1$ | $45.6 \pm 2.9$ | $57.5 \pm 1.1$ | $51.8 \pm 0.8$ | $57.8 \pm 1.3$ |
| All-OOV (%) | $56.3 \pm 0.7$ | $41.9 \pm 4.4$ | $17.8 \pm 0.6$ | $21.0 \pm 0.2$ | $24.7 \pm 2.1$ |
| Pre-OOV (%) | $89.4 \pm 0.4$ | $56.1 \pm 1.9$ | $80.8 \pm 0.4$ | $90.3 \pm 0.4$ | $91.7 \pm 1.6$ |
| Post-OOV (%) | $78.9 \pm 0.4$ | $50.9 \pm 1.8$ | $82.0 \pm 0.9$ | $85.7 \pm 0.4$ | $87.3 \pm 1.0$ |
| Mid-OOV (%) | $71.1 \pm 0.9$ | $50.9 \pm 2.4$ | $81.7 \pm 0.6$ | $79.4 \pm 1.3$ | $75.4 \pm 3.9$ |

Table 5: Win rates under commands with OOV words.

*Mid-OOV* indicate that the policies can also perform well when some OOV words are inserted. In *Pre-OOV* and *Post-OOV* commands, the original sequences are kept, which outperform *Mid-OOV*. Particularly in *Pre-OOV* commands, the performances are near the same with *Oracle*.

### D.6 Clustering Analysis of Command Representations.

We generate commands using some templates and visualize their embeddings with t-SNE. The templates are listed as follows:

- build building units:
  {build | create | make} {blacksmith | stable | barrack | workshop | guard tower},
  {build | create | make} a {blacksmith | stable | barrack | workshop | guard tower},
  {build | create | make} another {blacksmith | stable | barrack | workshop | guard tower}.

- build army units:
  {build | create | make} {swordman | cavalry | spearman | archer | dragon | catapult},
  {build | create | make} a {swordman | cavalry | spearman | archer | dragon | catapult},
  {build | create | make} another {swordman | cavalry | spearman | archer | dragon | catapult},
  {build | create | make} 3 {swordman | cavalry | spearman | archer | dragon | catapult},
  {build | create | make} 5 {swordman | cavalry | spearman | archer | dragon | catapult},
  {build | create | make} 7 {swordman | cavalry | spearman | archer | dragon | catapult}.

- attack: attack, kill.

- scout: scout, scout the map, go around the map.

- mine: mine, {gather | collect | mine} {minerals | resources}.

As shown in Fig. 4, similar commands are encoded into similar embeddings. In addition, the commands that build building units are close to each other and far from the commands that build army units, which suggests that the LSTM module is able to learn the semantic meanings of different commands.

## E  Human Evaluation Details

### E.1  Ethics Statement

The experiments are permitted under our department committee.There is not any personally identifiable information or sensitive personally identifiable information involved in the experiment.

When conducting the human evaluation experiments, we informed the participants in advance of the purpose of the experiment and the time the experiment might take. All the participants are fully informed and participate voluntarily with signed confirmation. We have controlled each experiment to be completed within 1 hour.

We invite 28 undergrad students and 2 Ph.D. students under department permission to complete a two-stage study. We release this human evaluation as an optional project in an undergrad course. The students who are interested in this project can participate. The Ph.D. students are well paid as research assistants.

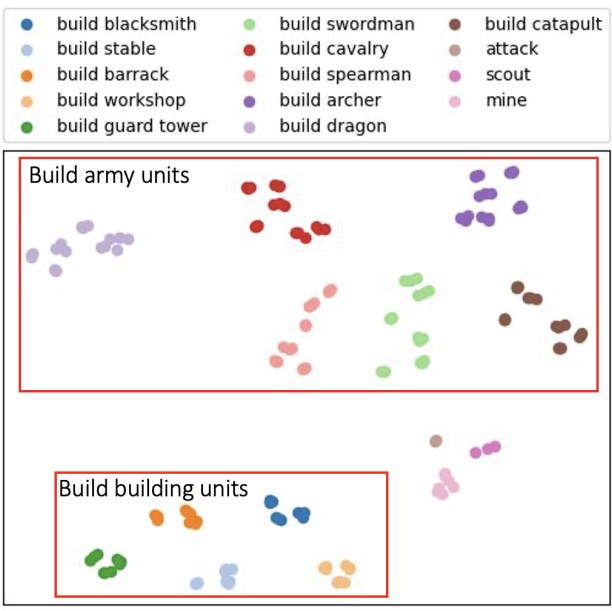

Figure 4: Visualization of command embeddings, similar commands are encoded into similar embeddings.

## E.2 Evaluation Process

We first introduce the rules of the game as described in Sec. A.1 and Fig. 1, and ask them to play the game by providing the commands. To avoid the participants not knowing what to do at the beginning, we give some basic commands as shown in Tab. 6. In addition to these basic commands, the participants can also input *arbitrary* commands. For example, some participants find that *"go around the map"* and *"scout the map"* can make the units explore the map and find the enemies.

| | |
|---|---|
| *mine* | *all peasant mine* |
| *attack* | *defend* |
| *build a blacksmith* | *build swordman* |
| *build a barrack* | *build spearman* |
| *build a stable* | *build cavalry* |
| *build a workshop* | *build archer* |
| *build dragon* | *build catapult* |
| *build a guard tower* | *build a town hall* |
| *build peasant* | |

Table 6: Basic commands presented to volunteers. The volunteers can also input *arbitrary* commands.

We pick 4 policies. i.e., RED, Switch, Joint and RL. In the first stage of the human evaluation, we shuffle the order of 4 policies and ask each volunteer to play with these policies using *arbitrary* commands. Once the volunteer feels familiar with the game policies, he/she is asked to rank the policies according to how the policies follow commands. In the second stage, each volunteer is asked to play 2 games per policy and try their best to win. In this stage, the volunteers can turn off the "fog-of-war" and observe the enemy units ("fog-of-war" is always turned on for the executor policies).

Note that we shuffle the policy order, and the students often spend a long time on the first policy to get familiar with MiniRTS, which can make the students biased towards the first policy they encountered. We think that a better evaluation process is to arrange an additional practice stage, allowing participants to play with a BC policy and get familiar with the game.

### E.3 Gameplay Interface

We present a screenshot of the gameplay interface in Fig. 5. The user can select a command from a set of recommendations using the "Select Command" button or input an arbitrary (English) command in the text box below. If the user types ENTER in the text box while leaving it empty, the previous command will be issued. This functionality is designed because we notice that it is a natural choice to keep sending the same command for several continuous time steps. To send an empty command (i.e., the NA command), the user can press the "Send Empty Command" button. Since some users may not be good at playing real-time strategy games, one can optionally toggle the fog of war using the "Fog of War: ON(OFF)". This lowers the difficulty of the game.

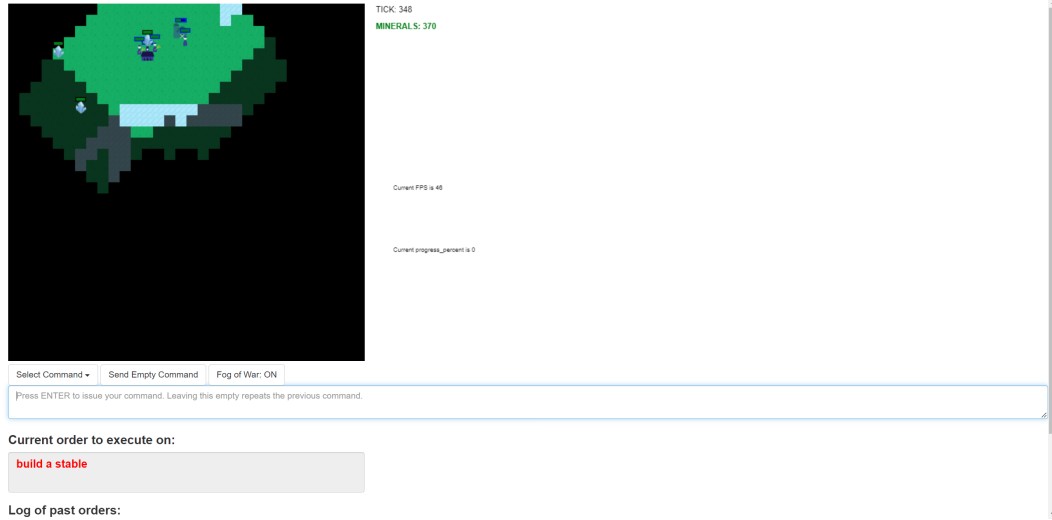

Figure 5: The gameplay interface.

### E.4 Emergent Behaviors

We find some interesting emergent behaviors during human evaluation. Please refer to our videos at https://sites.google.com/view/grounded-rl.

Some human participants prefer to build dragons since dragons sound powerful and cool. But building dragons makes it more challenging to win because dragons are expensive. If the player builds dragons at the beginning of a game, he/she will get few dragons ready when the enemies start to attack him/her, leading to a game loss. So the RL trained policies prefer to build "spearman", "swordman" or "cavalry", since win rates of these units are higher.

Fortunately, humans are intelligent and creative. Some participants figure out the behavior pattern of the built-in script AI. They find that the script AI will quickly build several army units and send them to attack. It is vital to resist the first two waves of attacks. A participant instructs the policy to construct several towers in the early stage of the game to resist the attack. After ensuring the safety, the student asks the policy to build dragons and then sends the dragons to find the enemies to attack, which finally results in a win.

It is difficult for pure RL to learn such a complicated strategy since building other army units (e.g., "spearman", "swordman" or "cavalry") is more direct for a fast win. A well-trained RED policy can follow human commands to execute this complicated strategy. In addition, although the human commands are vague, RED policy can take suitable actions to follow the commands. For example, it builds towers close to each other to strengthen the defense. After finding the enemies, it automatically sends the dragons to attack them.

Here we present the example of how the human commander instructs the RED policy to win a game using dragon by several commands in Fig. 6. There are 3 phases. In phase 1, the human player asks the policy to build towers by sending *"build a guard tower"*. In phase 2, the human player instructs the policy to build dragons using commands *"build a workshop"* and *"build dragon"*. In phase 3, the

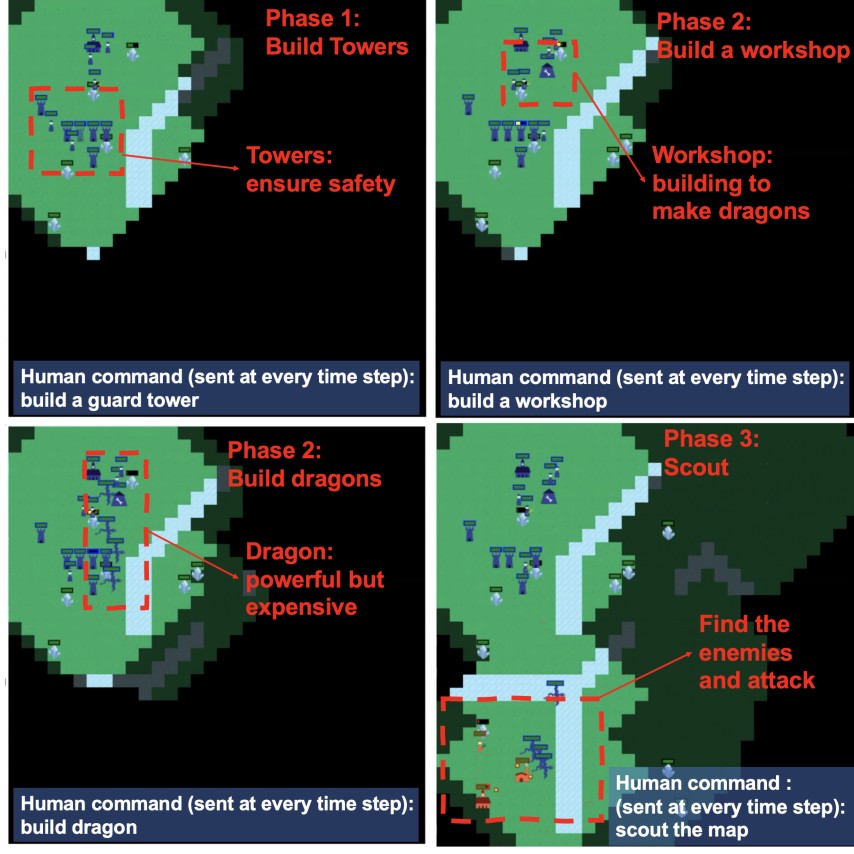

Figure 6: The overview of the "tower defense and dragon rush" strategy. The human player first asks the RED policy to build several guard towers to ensure safety, then build the dragons. The enemies try to attack when the RED policy is building dragons but are defended by the towers. After enough dragons are prepared, the human policy sends the command *"scout the map"*, then the dragons fly around the map, find and attack the enemies.

human player sends *"scout the map"*, and the policy makes the dragons fly around the map and find the enemies. We describe these 3 phases in detail in the following.

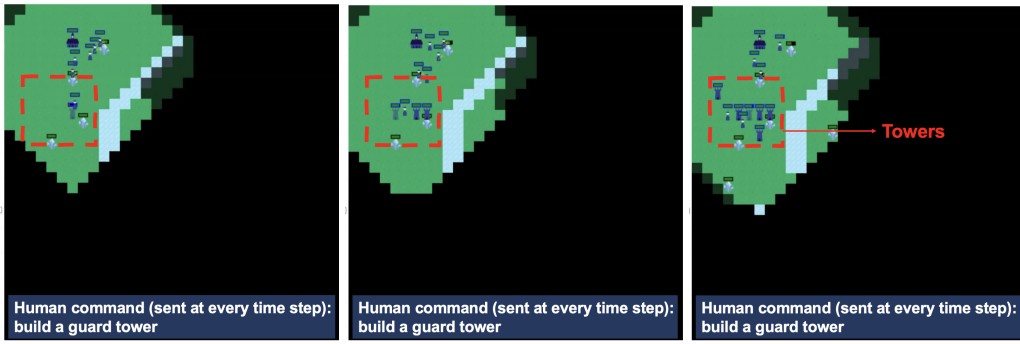

Figure 7: Phase 1: Build towers. The human player provides the command *"build a guard tower"* at every time step.

**Phase 1: Build Towers**  As shown in Fig. 7, in the early stage of the game, to ensure safety, the human player gives the command *"build a guard tower"* at every time step. So the RED policy lets one peasant build the tower, and the rest peasants keep mining resources from the resource units.

Since the command is provided at every time step, the peasant keeps constructing towers one after another.

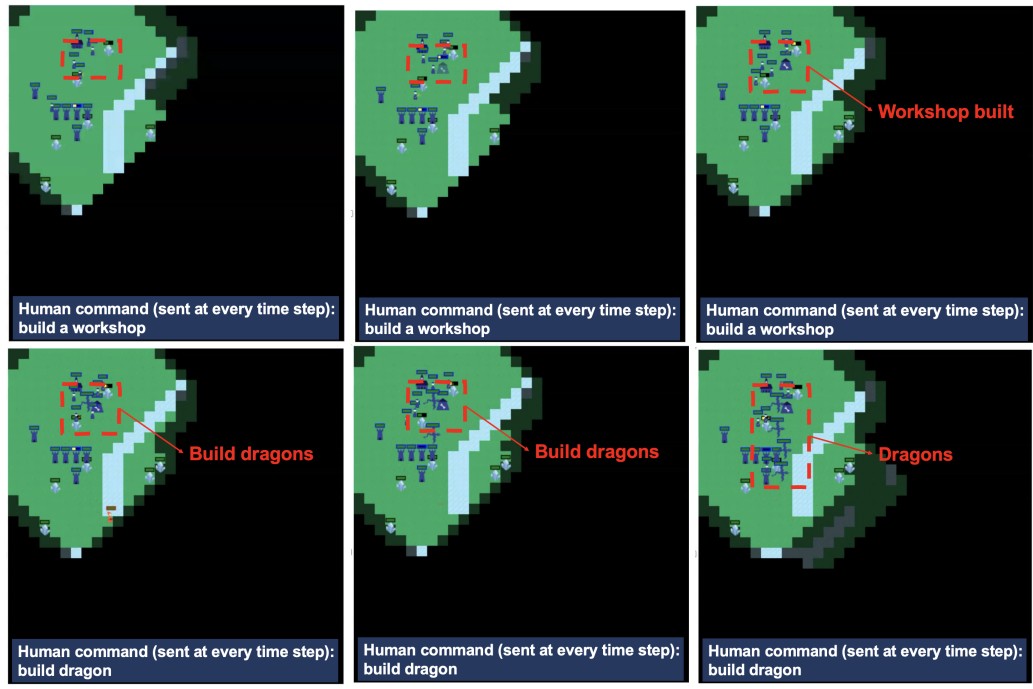

Figure 8: Phase 2: Build dragons. The human player first gives the command *"build a workshop"*, after the workshop is built, the human player changes the command to *"build dragon"*.

**Phase 2: Build Dragons**    As shown in Fig. 8, after building enough towers, the human player changes the command to *"build a workshop"*, where the workshop can produce dragons. Then one peasant starts to build the workshop immediately. Once the workshop is built, the human player changes the command and keeps asking the policy to *"build dragon"*. Then, the workshop keeps building the dragons, and all the peasants return to mine resources. In this phase, some enemy units started to attack but are all defended by the towers.

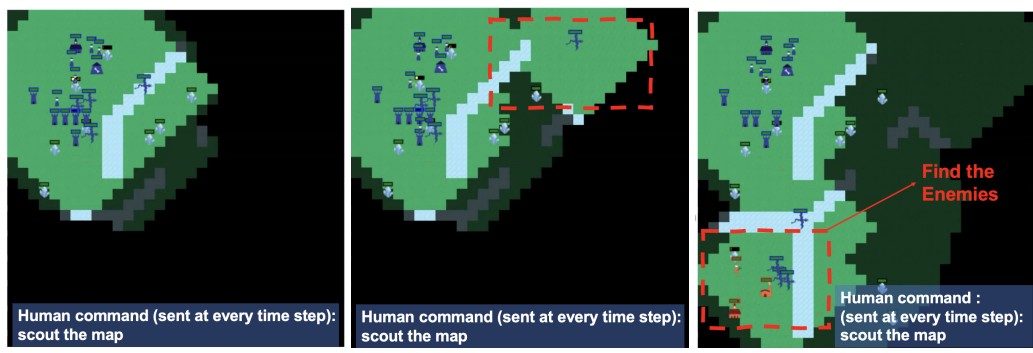

Figure 9: Phase 3: Find the enemies. The human player keeps giving the command *"scout the map"*.

**Phase 3: Find the Enemies**    As shown in Fig. 9, after building 4 dragons, the human player changes the command to *"scout the map"*, and the dragons fly around the map to explore. Once the dragons find the enemies, RED policy sends them to attack the enemies automatically.