# OpenReview forum: "Grounded Reinforcement Learning: Learning to Win the Game under Human Commands"
_NeurIPS.cc/2022/Conference — NeurIPS 2022 Accept_

### Official Review · Reviewer_wnXo · 2022-07-08

**Rating:** 6
**Confidence:** 4
**Soundness:** 3 good
**Presentation:** 3 good
**Contribution:** 3 good

**Summary:**

This paper takes on the challenge of both playing a game as optimally as possible, while following instructions and learning from demonstrations (both of which may be sub-optimal). The paper formulates this problem as a joint optimization, and tackles it by alternating steps of RL training with steps of distilling both demonstrations and successful RL trajectories. The paper evaluates this approach on the MiniRTS game, and evaluates against a range of baselines on various aspects such as rate of winning under different command types, how humans rank the policy’s instruction following, etc.

**Questions:**

See above, but briefly:
* Could the authors demonstrate more directly how well the agents follow commands—particularly in cases where they commands are strongly divergent from the winning policy?
* What do the authors think might be the risks of the proposed algorithm?

**Limitations:**

On the whole, I think this paper could be a useful contribution. The most important changes from my perspective would be (see above for full details):
* The claim that the learned policy is grounded is not particularly well supported. Experiments on how well the agents are actually following commands would be most important for improving my rating.
* There are risks to trying to minimally follow commands while pursuing another objective. Discussing these shouldn't be a major change.

Other limitations that I don't think are as critical to address, but would be helpful:
* Only evaluating on a single game makes it difficult to know how general the advantage is.

**Strengths And Weaknesses:**

Post-response update
-----------

The authors have addressed some of my main concerns, and more thoroughly evaluated their method. I believe these new experiments could be better integrated in to the main text, and the paper is still limited by experimenting only with a single environment, but I am increasing my score in accordance with the improvements.



Original review
--------------


Strengths:
* An interesting and well-motivated problem space and proposed methods.
* In-depth experiments evaluating different perspectives on the issues raised in one domain.
* Good ablations and baseline conditions.
* Generally well written and clear.

Weaknesses:
*The claim that the learned policy is grounded is not directly supported, only through circumstantial evidence that (while helpful) is not as direct as it could be.
  - The fact that win rates decrease with random commands is very weak evidence that the commands are being followed. The fact that humans rank the RED policy slightly higher than the rest is a little better, but the effect is weak (as noted in the paper there is an even distribution over the first rank). The statistical significance of the effect is not quantified, but there would almost certainly not be a significant difference between RED and the Switch or Joint baselines.
  - To address this concern, it would be ideal to actually evaluate how well the agent follows commands. For example, set up a situation like Fig. 1E, give the agent a command like “Retreat”, and then evaluate whether the agent actually moves its units away or simply continues attacking, and how long it maintains this behavior. This would be far stronger, direct evidence that the agent is actually following commands (rather than e.g. doing nothing, issuing resource gathering commands without actually retreating, etc.).
  - Performing these tests in *precisely* situations where the RL policy would do something very different would be the strongest test of behavior.
* Only evaluating on a single game makes it difficult to know whether the conclusions are generalizable; it would be ideal to perform experiments in 1-2 other settings as well.
* The general approach to the prior literature could be improved. For example:
  - The paper argues that “BC and IRL algorithms assume that the demonstrations are optimal”—however, a variety of prior works have addressed this problem in various ways, both from the perspective of IRL (e.g. https://proceedings.mlr.press/v97/brown19a.html) and from BC-like approaches conditioned on rewards (https://arxiv.org/abs/1912.02875), or just by using BC to initialize RL training (e.g., https://www.nature.com/articles/nature16961). While not all these perspectives would necessarily address the issue of also following commands, they are not incompatible with it, and the paper should discuss how such approaches might relate to (or complement) the proposed technique.
  - Relatedly, prior work on agents that follow human instructions (particularly https://arxiv.org/abs/2012.05672) has relied on a combination of BC, RL, and IRL to achieve better performance than BC alone; this would be useful to discuss.
* It seems to me that there is a fundamental tension between the goals of strictly following human commands, and winning the game. The proposed approach appears to pursue a policy that effectively tries to distort all human commands towards its own objective (i.e., to follow the command in the way that is most likely to lead to winning, as in Fig. 1C vs. 1D). Phrased in this way, it seems that there are clear alignment concerns about this approach. These should be discussed as a tradeoff/risk.
  - These concerns relate to the above questions about how well the agent is following commands—if the goal is for the agent to actually strictly follow human commands, it should not e.g. follow a retreat order for one second and then go right back to attacking.
  - It seems to me that it would be *far* safer to have a system that follows human commands first and foremost—i.e., a system that is “interruptable” (https://deepmindsafetyresearch.medium.com/building-safe-artificial-intelligence-52f5f75058f1 and https://www.ijcai.org/Proceedings/2017/0032.pdf)
  - This concern is *somewhat* ameliorated by the fact that the win rates under sub-optimal commands are much lower, but it depends on how frequently the commands are issued in these conditions.
  - Regardless, these issues should be discussed.

---

> ### Author Response · Authors · 2022-08-02
> **Response to Reviewer wnXo**
>
> ### Weakness 1: Could the authors demonstrate more directly how well the agents follow commands—particularly in cases where they commands are strongly divergent from the winning policy?
>
> We want to clarify that the RED, Joint, and Switch policies are all command-following policies. Theoretically, RED should not be substantially more obedient than Switch. It is because the Switch policy is just BC policy when not-NA commands are given while the demonstration is the only source for learning to follow commands. In practice, we believe that RED is slightly more preferred by human participants because it is much more stronger than BC policy, which makes humans easier to control the agents for a win.
>
> We also present a new metric that evaluates the command-following ability directly. That is, evaluating the ability to build an archer under the corresponding commands. More specifically, at the beginning of a game, we provide a sequence of commands to guide the policies to build archers. We then evaluate the success rates at the 30th time step. The results are listed in the following table.
>
> ||RL|IRL|Switch|Joint|RED|
> |:--:|:--:|:--:|:--:|:--:|:--:|
> |Success rate (%)|85.3 ± 1.2|2.2 ± 0.4|95.2 ± 0.4|96.4 ± 0.4|94.7 ± 1.8|
>
> We can observe that the Switch, Joint, and RED policies achieve high success rates, which suggests that they follow the commands well. RL and IRL polices tend to be command-ignorant and result in significantly lower success rates.
>
> It is also worth noting that RL achieves a high success rate. This is probably beacuse that building archers at the beginning of a game may be a relatively easy task. However, following commands in a complex situation later in the game may be more difficult. This means that the metric is likely to be insufficient.
>
> For the commands strongly divergent from the winning policy, we take reviewer 9kih’s suggestion. We implement the “adversarial oracle”, which always chooses the worst dominating units to build. The results are listed in the following table.
>
> ||RL|IRL|Switch|Joint|RED|
> |:--:|:--:|:--:|:--:|:--:|:--:|
> |Oracle (%)|89.3 ± 0.7|57.3 ± 3.9|78.7 ± 0.9|90.2 ± 0.7|92.6 ± 0.6|
> |NA (%)|57.5 ± 1.1|45.6 ± 2.9|57.5 ± 1.1|51.8 ± 0.8|57.8 ± 1.3|
> |Adv Oracle (%)|15.9 ± 0.7|35.0 ± 2.4|0.2 ± 0.1|0.4 ± 0.2|0.9 ± 0.4|
>
> We can observe that Switch, Joint, and RED policies obtain the winning rates of almost 0 under “adversarial oracle”. It suggests that they are command-following, even when the commands are strongly divergent from the winning policy.
>
> ***
>
> ### Weakness 2: It seems to me that there is a fundamental tension between the goals of strictly following human commands, and winning the game. The proposed approach appears to pursue a policy that effectively tries to distort all human commands towards its own objective (i.e., to follow the command in the way that is most likely to lead to winning, as in Fig. 1C vs. 1D). Phrased in this way, it seems that there are clear alignment concerns about this approach. These should be discussed as a tradeoff/risk.
>
> We sincerely appreciate the authors for pointing this out. Indeed, our mission is to build a humen-compatible AI, which should follow human commands as its first priority rather than environment rewards or win rates, so that we can have a safe AI. This motivates us to develop the concept of GRL. We have also briefly discussed this alignment issue in the introduction section (line 71), related work section (line 111) and conclusion section (line 383).
>
> Ideally, the system is interruptable by definition —- humans can always send a command like “retreat” or “stop” to a well-trained GRL agent to turn it off. In our experiment setting, the agent would not just retreat for 1 timestep and return to attack in the next timestep. This is because by default, our system will repeat the command from the previous timestep unless the human change it manually (in fact, the volunteer can simply press the enter button to repeat the previous command and humans like this feature). We have put more details on this in appendix E.3.
>
> Regarding the sub-optimal commands, we do find a lot of novel strategies invented by human volunteers —- including novel commands uncovered by the training data. These strategies are not necessarily the fastest for a win but they are indeed interesting. We have updated our website to include more examples.

---

> > ### Author Response · Authors · 2022-08-02
> > **Weakness 3: Related literature.**
> >
> >
> > ### Weakness 3: Related literature.
> >
> > We thank the reviewer for his suggestion on discussing more with related literature. Some works address the problem that human demonstrations are sub-optimal from different perspectives. ([https://proceedings.mlr.press/v97/brown19a.html](https://proceedings.mlr.press/v97/brown19a.html)) assumes the demonstrations are ranked and learns a reward function from these rankings. ([https://arxiv.org/abs/1912.02875](https://arxiv.org/abs/1912.02875)) proposes a BC-like approach by treating the rewards as inputs. ([https://www.nature.com/articles/nature16961](https://www.nature.com/articles/nature16961)) initializes RL training with a BC pre-trained model. These works concentrate on improving the performance, where the final policies are allowed to be **far from the demonstrations. **
> >
> > However, in our GRL setting, the policy needs to keep the knowledge of how humans follow the commands. It is much more challenging to improve the policy behavior while maintaining command-following ability, especially when the demonstrations are not optimal. Here we utilize an unconditioned policy to explore those states approaching a win and distill the generated trajectories and human demonstrations into a single neural policy, which improves the actions over those overlapping states.
> >
> > In addition, we want to clarify that the **commands** in the human demonstrations are often sub-optimal. But in previous works like ([https://arxiv.org/abs/2012.05672](https://arxiv.org/abs/2012.05672)), the task or goal of an episode in human demonstrations is defined by the language, causing the language commands are always oracle.

---

> > > ### Comment · Reviewer_wnXo · 2022-08-06
> > > **Thanks for the updates**
> > >
> > > Thanks to the authors for engaging with the review and updating the paper; I do find the added experiments and clarification have improved the contribution, although references to the supplemental experiments could be better integrated into the main text. I still find the paper to be somewhat limited by restricting experiments to a single game, but this is an understandable limitation and the paper is certainly improved; I will update my score accordingly.

---

### Official Review · Reviewer_WZeZ · 2022-07-09

**Rating:** 7
**Confidence:** 4
**Soundness:** 3 good
**Presentation:** 4 excellent
**Contribution:** 3 good

**Summary:**

Paper introduces a new task called “Grounded Reinforcement Learning” in which agents must be reward-maximizing with respect to some high-level language “constraint” (given by another agent/human). The authors demonstrate that existing vanilla RL, constrained RL (CRL), and behavior cloning (BC) fail to perform well both with and without commands in the MiniRTS environment (an example of this problem setup).

To solve this problem setting, the authors propose a new method RED - Reinforced Demonstration Distillation, which repeatedly alternates between pure RL and BC updates.  The authors demonstrate that RED is both the strongest and most grounded policy. The authors provide, through ablations, compelling arguments.

**Questions:**

1) It is unclear how this approach is different from having standard RL with a combined state space over both command and game state.  As the command representation is not learnt - it seems unclear how this actually is different to a standard RL problem.

2) Can more justification be given to how difficult the opponent you play with is? Can you play with a harder agent to more clearly demonstrate the advantages of your method?

3) It is really unclear how the command token is meaningfully different to having an additional dimension within the game state. Can some justification be given to how either this representation is learnt or has any assemblance of language (is it compositional, have semantic or pragmatic properties).


**Limitations:**

Addressing the above questions would be useful.

I also believe actually using a language model would take great steps in demonstrating the effectiveness of your approach.


**Strengths And Weaknesses:**

Strengths:

Introduces (somewhat) of a new problem statement

Clearly written

Human Evaluation is compelling

Impactful work from a safety perspective.

Weaknesses:

My largest criticism is the use of the word "language" and i think this undermines the entire approach - the current representation treats commands as one-hot representations. These are not learnt, shaped or any different to the state space. This has no standard components which i would consider natural language, such as pragmatics, syntax and semantics.

RED algorithm does not seem novel (effectively mixture of two)

Results are not significantly better than baselines.

---

> ### Author Response · Authors · 2022-08-02
> **Response to Reviewer WZeZ**
>
> ### Weakness 1: My largest criticism is the use of the word "language" and i think this undermines the entire approach - the current representation treats commands as one-hot representations.
>
> Please refer to our common response. **We sincerely apologize for our writing mistakes. We indeed focus on language commands rather than a fixed command set**. We adopt an LSTM encoder to encode arbitrary natural language commands, which is trained on 38,558 different commands from the training set. For human evaluation, the participants can issue arbitrary commands to cooperate with the policies. And we find that the RED policy can follow some novel instructions unseen during training.
>
> ***
>
> ### Weakness 2: Results are not significantly better than baselines.
>
> We have updated Tab.1 to make our results more clear. RED indeed outperforms baselines a lot.
>
> According to the results of Tab.1 and Fig.2, we observe that RL and IRL policies are weak-following, while RED, Switch, and Joint policies are strong-following. In Tab.1, when compared with weak-following policies, we need to focus on the “oracle”. RED outperforms the weak-following policies largely. When compared with strong-following policies, RED also performs the best.
>
> In addition, it is worth mentioning that the human evaluation also indicates that RED outperforms other baselines significantly. The participants think RED, Switch and Joint are all command-following policies (Fig.5), and the RED policy obtains the highest win rate (61.7%), which outperforms Joint (26.7%) and Switch (33.3%) by a clear margin (Fig.6).
>
> ***
>
> ### Weakness 3: RED algorithm does not seem novel. (effectively mixture of two)
>
> We want to emphasize that the problem of grounded RL is itself a novel and important problem rarely studied in the literature. Even though RED is technically simple, it is the first working algorithm on this novel problem and the whole algorithmic design can be beneficial to the community. In addition, we also want to emphasize that it is critical to adopt the unconditioned policy (i.e., use NA input) to collect RL trajectories to make RED work, which is technically novel and unique for our GRL problem. Fig.3 shows that RED policies trained with Random Command and Human Proxy are much less obedient. The analysis in Fig.4 also provides insights on how RED works.
>
> ***
>
> ### Q1: It is unclear how this approach is different from having standard RL with a combined state space over both command and game state. As the command representation is not learnt - it seems unclear how this actually is different to a standard RL problem.
>
> Please refer to our common response. Our command representation is indeed learned. Regarding the difference to standard RL, a critical issue is that there is not trivial way to verify whether a high-level language command is accomplished or not —- we can only learn from __sub-optimal__ demonstrations. We need to follow the commands (according to demonstrations) but also needs to improve it. Standard RL problem only cares about the final reward without following high-level language commands.
>
> ***
>
> ### Q2: Can more justification be given to how difficult the opponent you play with is?
>
> The difficulty of playing with the opponent can be inferred through the following perspectives:
>
> 1. A pure BC model under human proxy obtains a win rate of merely 7.8%.
> 2. If we train the policy with RL from scratch, the RL-trained policy can’t win any game. (Notice that the “RL” baseline in our paper is initialized from the BC pre-trained policy.)
>
> ***
>
> ### Q3: Can some justification be given to how either this representation is learnt or has any assemblance of language?
>
> We adopt an LSTM encoder to encode arbitrary natural language commands into fixed-length sentence embeddings. During the training process, there are a total of 38,558 different commands in the training set, and the parameters of LSTM are trainable. We also observe the RED policy can follow some commands unseen during training. Some examples have been presented on our website.

---

> > ### Comment · Reviewer_WZeZ · 2022-08-06
> > **Good Improvement!**
> >
> > Hey Authors!
> >
> > So it *massively* improves the paper that commands are provided as natural language and encoded via an LSTM into sentence embeddings!
> >
> > I think however, if all reviewers were confused by this, it highlights a flaw in the structure of the paper. Even if the sentence is removed or not specified theres very little analysis on the language used.  I'd expect a lot of analysis similar to when we began sequence modelling in NLP ([1,2], which evaluates how these agents perform on:
> >
> > * Out-of-Distribution Vocabulary
> > * Noisy Commands (this is different to random commands where an agent learns to ignore all)
> > * Incorrect commands.
> > * Clustering analysis of command (sentence) representations
> >
> > The problem setting is still exceptionally important and I think this is experiment protocol is useful for the community.  I will be revising my score based on the improvements presented.
> >
> > [1]- Mikolov, Tomas; et al. (2013). "Efficient Estimation of Word Representations in Vector Space"
> >
> > [2] - Bowman, Samuel R., et al. "Generating sentences from a continuous space." arXiv preprint arXiv:1511.06349 (2015).

---

> > > ### Author Response · Authors · 2022-08-08
> > > **Additional Results on noisy commands and the learned representation of the LSTM encoder**
> > >
> > > Thanks for your suggestions! These suggestions are indeed helpful to our paper!
> > >
> > > We have conducted additional experiments as suggested in the previous thread. We promise to include these new results in the final draft.
> > >
> > > ***
> > >
> > > ### Performance on noisy commands.
> > >
> > > We evaluate how the agents perform on noisy commands.
> > >
> > > We try 4 different ways of adding noise to **oracle** commands:
> > >
> > > - drop: We delete 50% of words in each command.
> > > - replace: We replace 50% of words in each command with random words (from the vocabulary).
> > > - insert: We insert random words (from the vocabulary) into each command, and make the command twice longer.
> > > - shuffle: We shuffle the words in each command.
> > >
> > > And the results are listed as follows.
> > >
> > > ||oracle|random|drop|replace|insert|shuffle|
> > > |:--:|:--:|:--:|:--:|:--:|:--:|:--:|
> > > |Strong-following|||||||
> > > |RED|92.6|29.8|61.1|59.2|77.17|77.3|
> > > |Joint|90.2|32.3|59.8|51.8|83.3|78.3|
> > > |Switch|78.7|12.3|39.6|26.7|77.8|66.9|
> > > |Weak-following|||||||
> > > |RL|89.3|53.3|60.5|57.8|75.5|72.7|
> > > |IRL|57.3|47.9|51.7|49.1|55.8|52.7|
> > >
> > > We can observe that the agents under noisy oracle commands perform better than random commands but worse than oracle. Intuitively, we can observe that the win rate roughly follows “random” < “drop”\~”replace” < “shuffle”\~”insert” < “oracle”. This suggests that the LSTM encoder is sensitive to input words and word order.
> > >
> > > We also find that strong-following policies are affected more by noisy commands than weak-following policies, indicating that strong-following policies are more sensitive to the commands.
> > >
> > > ***
> > >
> > > ### Performance on commands with Out-of-Vocabulary (OOV) words
> > >
> > > We also evaluate the policy performances on commands with OOV words. We use a special token [unk] to handle all the OOV words. We reuse the vocabulary from the [original MiniRTS paper](https://arxiv.org/pdf/1906.00744.pdf), where some rare words in the dataset are replaced with a special token “[unk]” during training. We insert the special token in different positions of the **oracle** commands to measure the influence of OOV words.
> > >
> > > - oov-only: We replace all command words with [unk] tokens.
> > > - pre-oov: We insert a sequence of [unk] tokens in front of each command, which makes the command twice longer.
> > > - post-oov: We insert a sequence of [unk] tokens at the end of each command, which makes the command twice longer.
> > > - mid-oov: We insert [unk] tokens between any two words in each command.
> > >
> > > And the results are listed in the following table.
> > >
> > > ||oracle|NA|oov-only|pre-oov|post-oov|mid-oov|
> > > |:--:|:--:|:--:|:--:|:--:|:--:|:--:|
> > > |Strong-following|||||||
> > > |RED|92.6|57.8|25.4|92.3|87.3|71.8|
> > > |Joint|90.2|51.8|21.2|89.9|86.3|80.1|
> > > |Switch|78.7|57.5|17.1|81.3|81.2|81.4|
> > > |Weak-following|||||||
> > > |RL|89.3|57.5|55.9|88.9|79.2|72.4|
> > > |IRL|57.3|45.6|47.9|58.7|52.7|54.3|
> > >
> > > The results show that “oov-only” commands substantially influence the performance of strong following policies (i.e., RED, Joint, and Switch), leading to much lower win rates than NA commands. For weak-following policies, RL and IRL policies under “oov-only” commands obtain similar win rates to NA.
> > >
> > > When oracle commands are given, inserting OOV words leads to a mild performance drop. The results of “pre-oov”, “post-oov” and “mid-oov” suggest that the LSTM encoder has learned how to handle the [unk] token.
> > >
> > > ***
> > >
> > > ### Performance on Incorrect commands.
> > >
> > > For incorrect commands, we take reviewer 9kih’s suggestion. We implement the **“adversarial oracle”** , which always choose the worst dominating units to build. The results are listed as follows.
> > >
> > > ||RL|IRL|Switch|Joint|RED|
> > > |:--:|:--:|:--:|:--:|:--:|:--:|
> > > ||Weak-following|Weak-following|Strong-following|Strong-following|Strong-following|
> > > |Oracle (%)|89.3 ± 0.7|57.3 ± 3.9|78.7 ± 0.9|90.2 ± 0.7|92.6 ± 0.6|
> > > |NA (%)|57.5 ± 1.1|45.6 ± 2.9|57.5 ± 1.1|51.8 ± 0.8|57.8 ± 1.3|
> > > |Adv Oracle (%)|15.9 ± 0.7|35.0 ± 2.4|0.2 ± 0.1|0.4 ± 0.2|0.9 ± 0.4|
> > >
> > > We observe that Switch, Joint and RED policies obtain win rates of almost 0 under “adversarial oracle”. It suggests that they are command-following, even when the commands are “incorrect”.
> > >
> > > ***
> > >
> > > ### Clustering analysis of command representations.
> > >
> > > We generate commands using some templates and visualize their embeddings with t-SNE. The [figure](https://drive.google.com/file/d/1LIgMiY0u8i9wlCz-kfXxlSk80p6ptLOy/view?usp=sharing) shows that similar commands are encoded into similar embeddings.
> > > In addition, the commands that build **building** units are close to each other and far from the commands that build **army** units, which suggests that the LSTM module is able to learn the semantic meanings of different commands.

---

> ### Comment · Reviewer_WZeZ · 2022-08-07
> **Update scores**
>
> Thanks for the comments -> I am update my score to a 7.
>
> I still think my suggestions would be make this a much more impactful paper / useful to the NLP community.

---

### Official Review · Reviewer_9kih · 2022-07-11

**Rating:** 5
**Confidence:** 3
**Soundness:** 2 fair
**Presentation:** 3 good
**Contribution:** 3 good

**Summary:**

This paper studies Grounded RL, where an agent is trained to obey natural language commands while still trying to attain the highest reward. The paper proposed to iterate between a pure RL loss and a behavior cloning loss, where the BC loss takes trajectories from both human demo and from the winning ones in the current on-policy sample queue.

The method is evaluated on the MiniRTS env and dataset and it is shown that the method can balance between high winning rate when no command is given, and low winning rate when the command is random.


**Questions:**

* Did you run any experiments to see whether the agent is able to follow certain commands with clear goals? E.g. “build two archers” is a clearly defined one and easy to evaluate if supported by the MiniRTS api.
* How did the “switch” policy do in terms of obeying the commands, especially the not-NA commands? It would be good to compare it with that of RED, to show that both can obey commands, but RED wins more – which is a central goal you are trying to achieve, iiuc.

Comments:
* It will be interesting to see an “adversarial oracle” where they always choose the worst dominating units to build.
* It is interesting that “human proxy” almost always performs worse than random.

**Limitations:**

The ratio between ||D_k|| and ||D|| and the threshold delta are dataset/env specific hparams which require extra evaluation to tune, and which will affect the command-obeying ability.

**Strengths And Weaknesses:**

Strengths:
* I appreciate the discussion around why the Lagrange multiplier is not suitable.
* The “random commands” is a good sanity check to see if the agent is affected by bad commands.
* The assumptions and the alternatives are clearly stated (e.g. l151-152, l243-244, etc.)

Weaknesses:
* Although often observed in practice, as mentioned in l204-206, the fact that the policy can generalize at all beyond BC for any not-NA command relies on the architecture of the policy network. If that is the core, then this paper did not study the condition for which such generalization occurs. And it’s such a core assumption of the proposed method, that to leave it understudied makes me a bit uneasy.
* While I appreciate the social impact section, “should not result in any negative social impact” is a bit too extreme. One can easily imagine military uses where humans provide data in a simulator and the policy is later deployed in the real world. A stronger command-following RL agent will have a negative impact there, allowing better performance compared to an RL agent trained from scratch (which doesn’t work in the MiniRTS case). You may be able to argue that it has no worse social impact than say BC or other RL methods, but saying “any” is too strong.

---

> ### Author Response · Authors · 2022-08-02
> **Response to Reviewer 9kih**
>
> ### Weakness 1:  The policy can generalize at all beyond BC for any not-NA command relies on the architecture of the policy network.  If that is the core, then this paper did not study the condition for which such generalization occurs.
>
> As for the architecture, the command embedding is concatenated with the observation embedding before outputting the action, which is a very standard implementation for a conditional policy. The LSTM encoding is able to encode different natural languages into a good embedding space for generalization, which is also typical in language grounding literature.
>
> We also remark that the demonstrations are highly sub-optimal (a pure BC only leads to a 7.8% win rate). So we use RED to fine-tune the BC pretrained policy to achieve a stronger policy that remains command following. The core technique is the RED training process rather than the architecture.
>
> ***
>
> ### Weakness 2: social impact
>
> Thanks for your suggestion. We have changed our argument and claim that our work has no worse social impact than BC or other RL methods.
>
> ***
>
> ### Q1: Did you run any experiments to see whether the agent is able to follow certain commands with clear goals? E.g. “build two archers” is a clearly defined one and easy to evaluate if supported by the MiniRTS api.
>
> We add an experiment evaluating the ability of the policies to build an archer in a certain number of steps. More specifically, at the beginning of a game, we provide a sequence of commands to guide the policies to build archers. The success rates at the 30th time step are listed in the following table.
>
> ||RL|IRL|Switch|Joint|RED|
> |:--:|:--:|:--:|:--:|:--:|:--:|
> |Success rate (%)|85.3 ± 1.2|2.2 ± 0.4|95.2 ± 0.4|96.4 ± 0.4|94.7 ± 1.8|
>
> We can observe that the Switch, Joint, and RED policies achieve significantly higher success rates than RL and IRL. This suggests that Switch, Joint, and RED faithfully follow the commands, while RL and IRL tend to be command-ignorant.
>
> On the other hand, it is also worth noting that RL still achieves a reasonable success rate. This is probably caused by the fact that building archers at the beginning of a game is relatively easy. However, following commands in a complex situation later in the game may be more difficult. This means that the metric is likely to be insufficient.
>
> ***
>
> ### Q2: How did the “switch” policy do in terms of obeying the commands, especially the not-NA commands? It would be good to compare it with that of RED, to show that both can obey commands, but RED wins more – which is a central goal you are trying to achieve, iiuc.
>
> It is actually difficult to verify whether a certain command is completed, so we eventually rely on the human evaluation. During human evaluation, human volunteers rarely adopt NA command —- we provide a hotkey for humans: when they press “enter”, the interface will directly execute the previous command. The human studies suggest that the "RED" policy outperforms “Switch” with a clear margin in win rate, which suggests our goal is achieved.
>
> ***
>
> ### Comment 1: It will be interesting to see an “adversarial oracle” where they always choose the worst dominating units to build.
>
> Thanks for your talented suggestion! We implement the “adversarial oracle” and the results are listed in the following table:
>
> ||RL|IRL|Switch|Joint|RED|
> |:--:|:--:|:--:|:--:|:--:|:--:|
> |Oracle (%)|89.3 ± 0.7|57.3 ± 3.9|78.7 ± 0.9|90.2 ± 0.7|92.6 ± 0.6|
> |NA (%)|57.5 ± 1.1|45.6 ± 2.9|57.5 ± 1.1|51.8 ± 0.8|57.8 ± 1.3|
> |Adv Oracle (%)|15.9 ± 0.7|35.0 ± 2.4|0.2 ± 0.1|0.4 ± 0.2|0.9 ± 0.4|
>
> We can observe that Switch, Joint, and RED policies obtain winning rates of almost 0, while RL and IRL policies still obtain relatively higher winning rates. These results also indicate that RL and IRL policies are command-ignorant, Switch, Joint, and RED policies are command-following.
>
> ***
>
> ### Comment 2: It is interesting that “human proxy” almost always performs worse than random.
>
> We think it is because that learning a good human commander is difficult. It is because
>
> 1. The dataset was collected to demonstrate how humans follow the language commands rather than winning, and most of the commands are highly sub-optimal.
> 2. The dataset is limited in size for training a good command generator.

---

> > ### Comment · Reviewer_9kih · 2022-08-03
> > **Follow up questions and clarifications**
> >
> > Thank you authors for responding to my comments and for the clarification posted around language. I think the clarification makes quite a big difference. Before the clarification, I was puzzled why the policy network would generalize from NA to commands with one-hot encoding. Now it makes a bit more sense, but I still have a few questions.
> >
> > Around “Weakness 1”, I can kind of see your point of view and why it has worked in your experiments. What I mean by “generalize” is that during training, all the policy has seen is the [(state, NA) -> action that wins] and the [(state, command) -> action that follows the command] data distribution. But during inference, somehow it is able to blend the two together and produce something sensible, such as [(state, command) -> action that follows the command and still tries to win]. How that happens is what I am unclear about.
> > My point of view was that if you take the extreme case where you have one policy network for NA and another for the rest of the commands, then I would expect that the policy is not going to generalize at all. If you follow your original “incorrect” setup, with one shared policy network and one-hot encoding of the top 500 commands, my guess is that it might generalize, but not as well. In your standard setup, it generalizes, as your experiments have shown.
> >
> > From what I can tell, you are proposing that in order to get a grounded (command following) agent that tries to win, all you need to do is to have a dataset annotated with commands, and access to the environment. Then you train an RL agent by doing 50% BC and 50% RL. If that is indeed all you need, then it is quite a powerful and general method. And because it seems quite general and powerful, I would like to understand when this “generalization” would happen.
> >
> > One perhaps related question: 1. In your answer to Q4 of oT9Z, you said “ Even though the RL baseline is never trained with non-empty commands during fine-tuning, we surprisingly find that the RL policy still keeps a weak command-following ability.”. That to me is quite strange. For the RL baseline, where does the command embeddings come from? Is it just random or is it pretrained? If it is random, then it would make sense that it could be command-following or command-ignoring (or even disobeying) about 50/50 of the time – unless the model setup or the input has some inherent bias.
> >
> > Another unrelated question: In table 2 (ablation study on the ratio), I'm trying to see how 2:1 is different from 1:0.5, Do you only have half of the human data available to do BC when the ratio is 1:0.5, and the model will never see the other half of the human data?
> >
> > Comments on your reply to my other questions: thank you for running the additional experiments. Now I am more convinced that this method indeed works as expected.

---

> > > ### Author Response · Authors · 2022-08-04
> > > **Response to the follow up questions**
> > >
> > > **Thanks for your response and insightful questions.**
> > >
> > > ### Where do the command embeddings come from for the RL baseline?
> > >
> > > The RL baseline is initialized from the **pre-trained** BC policy. The BC policy learns from the paired demonstrations with an LSTM encoder. During RL fine-tuning, the LSTM encoder is never used but the final policy still has a weak command-following capability at test time. MiniRTS is so complex that pure RL from scratch fails to learn any meaningful winning strategies (0% win rate).
> > >
> > > ### Why and when does the policy generalize?
> > >
> > > It is indeed an interesting empirical finding that simply by repeating RL and BC and utilizing a unified conditional network, the RED policy can eventually generalize. We did attempt to analyze this observation, so an intuitive justification is provided in Sec.4.3 (lines 233 - 239) of our paper. Our hypothesis is as follows. The RL training leads the policy to frequently visit those states approaching a win. In the self-distillation phase, we distill two drift distributions into a single neural policy, so the actions over those overlapping states are more likely to be improved. We provide evidence for this hypothesis in Sec.5.2 (Fig.4). Please refer to our paper for more discussions.
> > >
> > > It is also worth noting that RL training with NA command is important. Fig.3 shows that training with random commands or human proxy makes the policy command-ignoring. This is because when a language command is provided, the environment reward optimized by RL will not be well aligned with the given command, which breaks the policy and makes it command-ignorant.
> > >
> > > ### The difference between 1:0.5 and 2:1.
> > >
> > > “X: Y” means that **in each iteration** (which contains many mini-batches), we __randomly__ pick the Y portion of demonstrations and the X portion of RL samples. So when Y is larger, each demonstration data will be learned more times by the network within the same amount of iterations, which makes the policy overfit more.
> > >
> > > So the motivation for comparing 1:0.5 and 2:1 is to examine what if we simply increase the frequency of learning each demonstration without changing the X: Y ratio. The results show that even if the ratio is unchanged, seeing demonstrations more frequently (by mini-batches) will make the network overfit more.
> > >
> > > We want to remark that even for 1:0.5, the network is able to distill all the demonstrations since we **resample** 50% of demons in each iteration.

---

### Official Review · Reviewer_8YRG · 2022-07-12

**Rating:** 5
**Confidence:** 3
**Soundness:** 2 fair
**Presentation:** 2 fair
**Contribution:** 2 fair

**Summary:**

This paper studies the problem of building reinforcement learning agents constrained by human demonstrations in the real-time strategy game MiniRTS.

**Questions:**

1. In the human study, how did the author evaluate the level of skills of participants and how they determined when the participants were "familiar" (line 353) with the game?
2. Could the authors clarify the difference between the framework GRL they proposed and existing language grounded approached? The manuscript would benefit from the authors addressing the difference of their setting with existing language grounded approaches. I do not believe that the complexity of MiniRTS as an environment is a sufficient argument to dismiss previous work.

**Limitations:**

The author address some limitations of their work and the setting discussed in the paper is interesting from its potential societal impact of having RL agent that trade-off following human commands with the optimality of the actions it executes.

**Strengths And Weaknesses:**

**Strengths** The paper proposes an interesting approach, REinforced demonstration Distillation (RED), to find RL agents that trade off following possibly suboptimal human commands with finding an optimal solution to the MiniRTS environment. I think this is an interesting setting, and the manuscript presents the problem and related work in a clear and concise manner. The baselines the authors selected are appropriate for the problem set up

**Weaknesses**
While the manuscript claims that MiniRTS is harder than following natural language instructions in some grid world navigation environments (e.g., [1]). For starters, the problem of training from natural language seems harder for two reasons. First, understanding natural language instructions seems a harder problem than the command setup proposed in the paper. Furthermore, the problem that this previous navigation work analyzes is usually under sparse rewards (success/fail) while in the setting proposed in the paper there are dense demonstrations that may alleviate the complexity of the task.

In terms of novelty, constraining RL with demonstrations as has been extensively studied in the literature [2,3] and the architectural challenges for solving the language-grounded task were introduced in MiniRTS [4].

[1] Chevalier-Boisvert et al. BabyAI: A Platform to Study the Sample Efficiency of Grounded Language Learning. 2018
[2] Yang et al. Accelerating Safe Reinforcement Learning with Constraint-mismatched Policies. 2020
[3] Goecks et al. Integrating Behavior Cloning and Reinforcement Learning for Improved Performance in Dense and Sparse Reward Environments
[4] Hu et al. Hierarchical Decision Making by Generating and Following Natural Language Instructions

---

> ### Author Response · Authors · 2022-08-02
> **Response to Reviewer 8YRG**
>
> ### Weakness 1: Understanding natural language instructions seems a harder problem than the command setup proposed in the paper.
>
> In our paper, the commands are indeed natural language instructions. A text sequence is encoded into a sentence embedding through an LSTM encoder. When evaluating the policy, arbitrary nature language could be sent as the command. During the human evaluation process, the participants are also allowed to provide arbitrary language commands. We find that the RED policy can also follow some instructions unseen when training, some demonstrations are presented on our website. Please also refer to the common response.
>
> ***
>
> ### Weakness 2: The problem that this previous navigation work analyzes is usually under sparse rewards (success/fail) while in the setting proposed in the paper there are dense demonstrations that may alleviate the complexity of the task.
>
> In many testbeds like [1],  there often exists an oracle that can verify whether a state is consistent with the language description, so the reward function associated with the command is convenient to obtain, and the RL algorithm can be applied directly. In our setting, verifying if a high-level command is completed is difficult, so the policies can only learn how humans follow natural language commands from the demonstrations.
> Regarding human demonstrations, we want emphasize that human behaviors are highly sub-optimal — a pure BC policy from human proxy achieves only a 7.8% win rate. The dataset is primarily used for learning how to __follow human commands__  rather than directly learning a strong winning strategy[4].
>
> ***
>
> ### Weakness 3: constraining RL with demonstrations as has been extensively studied in the literature [2,3], and the architectural challenges for solving the language-grounded task were introduced in MiniRTS [4].
>
> Our proposed GRL problem is indeed a novel problem, which is much more difficult than the existing language-grounded problem. In the GRL setting, we encounter the problem that 1) the reward associated with commands is unavailable, and 2) the demonstrations are not optimal.
>
> RED is inspired by many techniques [2,3], but there are still many differences. We have discussed the relationship between our work and constrained RL in the “related work” section, please refer to "Constrained reinforcement learning" for more details. We also have other technique contributions unique to our problem. That is, we train an **unconditioned** policy using RL for higher win rates and distilling two drift distributions into a single neural policy. Please refer to Sec.4.3 (lines 233 - 239) for more justification.
>
> ***
>
> ### Q1: In the human study, how did the author evaluate the level of skills of participants and how they determined when the participants were "familiar" (line 353) with the game?
>
> This is a subjective judgment by the participants. If the participants feel confident to distinguish and rank different policies, we consider them familiar with the game.
>
> ***
>
> ### Q2: Could the authors clarify the difference between the framework GRL they proposed and existing language grounded approached?
>
> There are two main differences between the GRL and the existing language-grounded settings. We have discussed the differences in the “related work” section, please refer to “Language grounding” for more details.
>
> 1. Many testbeds adopt template-based languages over objects (e.g., box or cube) and attributes (e.g., spatial relation or color),  and **there exists an oracle that can verify whether a state is consistent with the language description**. Then the reward function associated with the command is convenient to obtain, and the RL algorithm could be adopted directly. However, GRL focuses on high-level natural commands.**It is difficult to verify if a command is completed**, so we can only learn how humans follow natural language commands from the demonstrations.
> 2. Recent visual navigation benchmarks provide large-scale human demonstrations. In these settings, **every human description is paired with a successful trajectory,** so the demonstrations are actually optimal. But in the GRL setting, **both commands and game trajectories are highly sub-optimal.**
>
> [1] Chevalier-Boisvert et al. BabyAI: A Platform to Study the Sample Efficiency of Grounded Language Learning. 2018
>
> [2] Yang et al. Accelerating Safe Reinforcement Learning with Constraint-mismatched Policies. 2020
>
> [3] Goecks et al. Integrating Behavior Cloning and Reinforcement Learning for Improved Performance in Dense and Sparse Reward Environments
>
> [4] Hu et al. Hierarchical Decision Making by Generating and Following Natural Language Instructions

---

### Official Review · Reviewer_oT9Z · 2022-07-13

**Rating:** 6
**Confidence:** 4
**Soundness:** 3 good
**Presentation:** 3 good
**Contribution:** 4 excellent

**Summary:**

This work focuses on grounded reinforcement learning. Generally, it means finding a conditional RL policy that can perfectly response to high-level command.

In this paper, the authors proposed a constraned RL algorithm RED to (1) train an agent that is capable of winning strategy games and (2) follow most human instructions. Concretely, RED uses RL to learn explorative policy while uses BC to imitate human demonstration.

Experiment results suggest RED achieves command-following policies and higher winning rate.

**Questions:**

### Questions on method


Sec 4.3 says this work uses Lagrangian multiplier to solve the constrained optimization problem in Eq2. It turns out that, in my understanding, this is simply a weighted loss. A formal Lagrangian method should contain update process of the $\beta$, which is just ignored by this paper (or I miss, please point out). The later "iterative solution" also breaks the so called "Lagrangian method".

---

Line 240 and Appendix A.2: Could you discuss more details on how you deal with action space? What does the "0/1 action for each unit" means? Who does the selection?

---

How to insert a NA command? IIUC, there exist a fixed set of possible commands right? So NA command should be just one more command in that set. You create embedding for each command in the set and than pass the embedding to the neural network.

---





### Questions on experiments

I can't understand table 1. IIUC, these four baselines have the policies that do not take command as input. Why they have different performance in Table 1 when applying different test commands? Please describe how you turn them as conditional policies.

For example, since RL is trained without any objective to conduct distinguishable behaviors under different commands, why inserting different test commands will lead to different behavior?

---

I don't think Table 1 is convincing, especially when taking STD into account.

Line 288: "Both RED and Joint policies substantially outperform Switch (BC) policy with sub-optimal commands (Random and Human Proxy), suggesting the conditioned behaviors are improved." This is a overclaimed statement. I can't draw this conclusion from Table 1.

---

I like Fig 2, which shows the dependency between commands and behaviors. Many "conditional RL" methods fail to prove that the conditional policy has casuality to the input command. This figure also shows that RED has much better "casuality" than RL.

My problem is that, in Figure 2, RL in 0% random commands achieves ~57% win rate. Does this value align with the RL performance with NA test command in Table 1?

Again, if RL is trained compelety without human command, then (1) where you can insert the command to the policy? and (2) why RL performance changes with the percentage of random commands?

---


Figure 6 says RL policy achieves similar win rate compared to RED. IIUC, RL policies are not conditioned on any human command. This result suggest that a human subject cooperates with RED trained conditional policy will lead to RL-comparable performance? This seems to be a violation to Table 1 "Oracle" result, which might lead to

---

There is only one real gameplay footage provided (less than 2 minutes), with only 6 commands given, which is far less than convincing. Meanwhile, the commands seemed to be programmed/copy-pasted (I did not see any input lag), is there any real human intervention involved in recoding this video?

---

This game seems to be very complicated. Does your first intention of the application of this algorithm to be this high-level game? I understand the miniRTS environment is the only one that fitted the algorithm's need, but if you have alternatives (even if the performance is weak), would you also like to present them in the paper? Having diversity is a huge plus to your paper.








### Questions on paper




### Other questions or concerns



The [google drive link](https://drive.google.com/file/d/129ZV9CU6zUnp5B4v0p91Xaae8-M5zxHy/view) provided is **not anonymous**. This is a violation of the NeurIPS Code of Conduct (double-blind reviewing policy). Please action ASAP.

---

This paper has experiments involving 30 college students experiments, are attendees aware of their rights and responsibilities? A more explicit Ethics Statement should be included.

---

Conditional BC is not a new idea. Please refer to this work: End-to-end Driving via Conditional Imitation Learning

---

Appendix D, as said in the checklist, has no information on human experiment. You moved them to Appendix E.

---

The website given in the appendix is very informative and I like the real-time gameplay footage, but there are several problems I want to point out:
- Code link is broken.
- Many grammar errors, will not be present here (plural or singular, tense, etc)


**Limitations:**

The authors provide limitation section.

GRL is an interesting field. How to improve human-AI cooperation in test-time, how to learn a better human proxy, and how to further encourage, illustrate, quantify the causity between command and behaviors might be future directions.

**Strengths And Weaknesses:**


### Strengths

* The idea is concise. Use BC to learn human behavior while use RL to explore better solution.
* Visualization and videos provided are mostly clear and contain necessary information.
* Fig1 is illustrative and helpful to explain what is grounded reinforcement learning. (It would be better if Fig 1 has no background color and in PDF format)
* Code is released.

### Weaknesses

* This work assumes reward from interactive environment. (limitation)
* The code has no detailed guidance and documentation.
* Though I like Figure 2 and other experiments, the main experiment results in Table 1 is confusing and non-convincing.
* The idea is not novel. Conditional imitation learning is a long lasting topics. Refer to "End-to-end Driving via Conditional Imitation Learning"
* Many details are missing or confusing, such as Table 1 and Lagrangian method.

---

> ### Author Response · Authors · 2022-08-02
> **Response to Reviewer oT9Z**
>
> We thank the reviewer for carefully reading, constructive suggestions, and helpful questions. We sincerely apologize that our presentation is confusing somewhere. We have revised our draft to make it more clear.
>
> ### Q1: Lagrangian multiplier?
>
> Thanks for the reminder that the $\beta$ coefficient should be adaptive for a theoretical convergence guarantee. We remark that using a fixed $\beta$ coefficient follows a popular implementation practice in recent deep RL literature for its simplicity and training stability [[https://arxiv.org/abs/2104.02180](https://arxiv.org/abs/2104.02180), [https://arxiv.org/abs/2205.01906](https://arxiv.org/abs/2205.01906)]. Our RED method is motivated by the recent empirical successes of adopting supervised learning [[https://arxiv.org/abs/2206.12030](https://arxiv.org/abs/2206.12030), [https://arxiv.org/abs/2010.02975](https://arxiv.org/abs/2010.02975)] techniques in RL. We also additionally conduct experiments with an adaptive $\beta$ with careful parameter tuning. It is worth noting that such a technique can also be applied to our proposed method RED as well. In particular, during the BC phase, we add a coefficient $\beta$ on the behavior clone loss on demonstrations. This coefficient $\beta$ is updated similarly during the training process. The results are listed in the following table (also in appendix D.2):
>
> ||Switch|Joint (fix $\beta$)|RED (fix $\beta$)|Joint (Adap. $\beta$)|RED (Adap. $\beta$)|
> |:--:|:--:|:--:|:--:|:--:|:--:|
> |Oracle (%)|78.7 ± 0.9|90.2 ± 0.7|92.6 ± 0.6|93.7|93.2|
> |NA (%)|57.5 ± 1.1|51.8 ± 0.8|57.8 ± 1.3|57.3|61.6|
> |||||||
> |Random (%)|12.3 ± 0.2|32.3 ± 0.3|29.8 ± 0.8|36.7|36.2|
>
> Indeed, an adaptive $\beta$ leads to better performances for both RED and Joint methods. When having an adaptive $\beta$, RED achieves a comparable performance with Joint while still outperforming all the baselines under the NA command.
>
> ***
>
> ### Q2: Could you discuss more details about the action space? What does the "0/1 action for each unit" mean? Who does the selection?
>
> In the original MiniRTS environment, at each timestep, different actions are specified for every controlled unit, which is too high-dimensional for RL training (so the original MiniRTS paper only conducts supervised training). Our paper simplifices the action space by first selecting which units to control and then specifying a shared action for all the selected units, which works well in practice. More specifically, our policy network outputs:
>
> 1. A **common** action;
> 2. A 0/1 flag for each unit.
>
> Then we convert this into the standard MiniRTS format in the following way:
>
> 1. For a unit assigned “1”, it is selected by the policy and will execute the common output action;
> 2. For a unit assigned “0”, it is not selected and will execute the action CONTINUE (repeat the action of last timestep).
>
> Such a conversion process makes our output actions compatible with the MiniRTS API.
>
> ***
>
> ### Q3: How to insert a NA command?
>
> In our implementation, the NA command corresponds to an empty sequence. Since our model encodes the language command into a fixed-size embedding by an LSTM encoder, the embedding of an empty sequence (i.e. the NA command) is simply a zero vector. Please refer to the common response for our clarification on our language encoder module.
>
> ***
>
> ### Q4: Why do the baselines have different performance in Table 1 when applying different test commands?
>
> __All__ the policy networks are conditioned on __both__ the game observation and **an command embedding**, which is processed by the LSTM module from the (possibly empty) input text sequence. So, different policies, which are trained by different algorithms, will output different actions when different commands are given.
>
> Even for the “RL” baseline, it is still a conditioned policy. For the RL baseline, we tune the conditional BC policy with PPO with NA as its input command. Even though the RL baseline is never trained with non-empty commands during fine-tuning, we surprisingly find that the RL policy still keeps a weak command-following ability. When evaluating the RL policy, we feed different meaningful commands instead of the NA command and observe that the RL policy behaves differently under different commands.

---

> > ### Author Response · Authors · 2022-08-02
> > **Q5 - Q10**
> >
> > ### Q5：Line 288: "Both RED and Joint policies substantially outperform Switch (BC) policy with sub-optimal commands (Random and Human Proxy), suggesting the conditioned behaviors are improved." This is an overclaimed statement. I can't draw this conclusion from Table 1.
> >
> > We realize that the original Tab.1 may be hard to interpret. We have updated the table in our revision. We want to clarify that
> >
> > 1. RL and IRL policies (i.e. the first two columns) are less obedient. Even under random commands, their win rates remain comparable to the NA case. Switch, Joint, and RED policies (i.e. the last three columns) are obedient and perform much worse under inappropriate commands.
> > 2. When given oracle commands (i.e. the first row), RED  is the highest.
> > 3. When given NA command (i.e. the second row), RL (57.5), Switch (57.5), and RED (57.8) policies are comparable, and they all outperform the Joint (51.8) policy.
> > 4. When given sub-optimal commands (i.e. the last two rows, Random and Human Proxy), the win rates of RED policy (29.8 & 11.0) and Joint policy (32.3 & 11.4) indeed substantially outperform Switch (BC) policy (12.3 & 7.8), suggesting the behaviors conditioned on sub-optimal commands are also improved __among all the command-following variants__.
> >
> > In addition, we also evaluate the performance of different methods under ``adversarial oracle’’ commands, which always choose the worst dominating units to build. The win rate following this adversarial oracle is expected to be close to 0. The results again suggests that RL and IRL policies are command-ignorant, while Switch, Joint, and RED policies are command-following.
> >
> > ||RL|IRL|Switch|Joint|RED|
> > |--|:--:|:--:|--|--|--|
> > ||*Command-ignorant*|*Command-ignorant*|**Comamand-following**|**Comamand-following**|**Comamand-following**|
> > |Oracle (%)|89.3 ± 0.7|57.3 ± 3.9|78.7 ± 0.9|90.2 ± 0.7|92.6 ± 0.6|
> > |NA (%)|57.5 ± 1.1|45.6 ± 2.9|57.5 ± 1.1|51.8 ± 0.8|57.8 ± 1.3|
> > |||||||
> > |Random (%)|53.3 ± 0.4|47.9 ± 2.8|12.3 ± 0.2|32.3 ± 0.3|29.8 ± 0.8|
> > |Human Proxy (%)|43.6 ± 0.6|48.9 ± 2.9|7.8 ± 0.1|11.4 ± 0.7|11.0 ± 0.4|
> > |||||||
> > |Adversarial Oracle (%)|15.9 ± 0.7|35.0 ± 2.4|0.2 ± 0.1|0.4 ± 0.2|0.9 ± 0.4|
> >
> > ***
> >
> > ### Q6: In Fig. 2, RL in 0% random commands achieves ~57% win rate. Does this value align with the RL performance with NA test command in Table 1?
> >
> > Yes. RL in 0% random commands aligns the RL performance with “NA” test command (57.5) in Tab.1, and RL in 100% random commands aligns the RL performance with the “Random” test command (53.3) in Tab.1.
> >
> > ***
> >
> > ### Q7: In Fig. 2, if RL is trained completely without human command, then (1) where you can insert the command to the policy? and (2) why RL performance changes with the percentage of random commands?
> >
> > We want to clarify that the RL policy is __fine-tuned__ from the pretrained BC policy (please refer to line 262). So it is still a language-conditioned policy even though it does not take any non-empty commands during RL fine-tuning.
> >
> > ***
> >
> > ### Q8: Figure 6 says RL policy achieves similar win rate compared to RED. IIUC, RL policies are not conditioned on any human command. This result suggest that a human subject cooperates with RED trained conditional policy will lead to RL-comparable performance? This seems to be a violation to Table 1 "Oracle" result, which might lead to
> >
> > The RL policy often ignores human commands for a win while RED consistently follows human commands. Note that human commands are often sub-optimal. So, it is often the case that after policy improvement, RED merely achieves comparable performances with RL under human commands. However, if the commands are optimal (e.g., Oracle), RED becomes superior.
> >
> > ***
> >
> > ### Q9: is there any real human intervention involved in recoding this video?
> >
> > The commands in this video are indeed issued by a human.The video is the replay of a recorded game instead of the actual game interface with input lag omitted for visualization purpose. We notice that it is a common choice to keep sending the same command for several continuous time steps. So, when the player presses “enter” button in the gameplay interface, a repeated command will be sent. In fact, most volunteers do like to use this feature. This is why the commands in the video seem to be copy-pasted.
> > We have presented the actual play interface in our new version of the appendix E.3.
> >
> > In the provided video, we want to show that by providing a small number (e.g. 6) of commands, one can lead the model to a set of novel behaviors not covered by the training demonstrations. We also present more gameplay videos with unseen language commands on the website.
> >
> > ***
> >
> > ### Q10:  Does your first intention of the application of this algorithm to be this high-level game?
> >
> > We found MiniRTS to be the best fit for the GRL problem. We were working on this environment all the time and found that the RED algorithm is a great solution.
> >
> > ***

---

> > > ### Author Response · Authors · 2022-08-02
> > > **Q11 - Q15**
> > >
> > > ### Q11: The google drive link provided is not anonymous. This is a violation of the NeurIPS Code of Conduct (double-blind reviewing policy). Please action ASAP.
> > >
> > > Thanks for pointing this out. It has been fixed.
> > >
> > > ***
> > >
> > > ### Q12: This paper has experiments involving 30 college students experiments, are attendees aware of their rights and responsibilities? A more explicit Ethics Statement should be included.
> > >
> > > The experiments are permitted under our department committee.There is not any personally identifiable information or sensitive personally identifiable information involved in the experiment.
> > >
> > > When conducting the human evaluation experiments, we informed the participants in advance of the purpose of the experiment and the time the experiment might take. All the participants are fully informed and participate voluntarily with signed confirmation. We have controlled each experiment to be completed within 1 hour.
> > >
> > > ***
> > >
> > > ### Q13: Conditional BC is not a new idea. Please refer to this work: End-to-end Driving via Conditional Imitation Learning.
> > >
> > > Conditional BC is indeed not a new idea, but the goal of our work is not just conditional BC. Taking the paper End-to-end Driving via Conditional Imitation Learning as an example, our work differs from this work for these reasons:
> > >
> > > 1. In our work, the human demonstrations are sub-optimal and are used only to demonstrate command-following. On the other hand, conditional BC typically assumes optimal demonstrations.
> > > 2. Our work takes natural language commands as inputs, so the number of possible input commands to our model can be exponentially large. We also observe that the policy can follow some commands unseen during the training stage.
> > >
> > > We have discussed the relationship between our work and conditional BC in the introduction section (lines 40 - 49). We also cite this work in our new version.
> > >
> > > ***
> > >
> > > ### Q14: Appendix D, as said in the checklist, has no information on human experiment. You moved them to Appendix E.
> > >
> > > Thanks for pointing this out. We have revised our paper accordingly.
> > >
> > > ***
> > >
> > > ### Q15: The website given in the appendix is very informative and I like the real-time gameplay footage, but there are several problems.
> > >
> > > We have fixed the code link, and carefully checked the grammar on the website. We also present some more videos on our website.

---

> > ### Author Response · Authors · 2022-08-09
> > **STD for the experiments of adaptive $\beta$.**
> >
> > We repeat the experiments of adaptive $\beta$ on 3 seeds. The results are listed in the following table, and the conclusions remain unchanged.
> >
> > ||Switch|Joint (fix $\beta$)|RED (fix $\beta$)|Joint (Adap. $\beta$)|RED (Adap. $\beta$)|
> > |:--:|:--:|:--:|:--:|:--:|:--:|
> > |Oracle (%)|78.7 ± 0.9|90.2 ± 0.7|92.6 ± 0.6|93.7 ± 0.1|93.1 ± 0.1|
> > |NA (%)|57.5 ± 1.1|51.8 ± 0.8|57.8 ± 1.3|57.7 ± 0.3|61.2 ± 0.3|
> > |||||||
> > |Random (%)|12.3 ± 0.2|32.3 ± 0.3|29.8 ± 0.8|38.1 ± 1.1|37.6 ± 1.2|

---

### Author Response · Authors · 2022-08-02
**We have updated our paper to address value comments from the reviewers. All the changes are highlighted in red.**

### Our paper does tackle natural language commands rather than a fixed command set. We have corrected the wrong presentation in our revised draft.

We sincerely apologize for our writing mistakes from line 150 to line 153 (old version). “Particular in the paper, we consider a finite set of top 500 commands … It is also possible to extend C to arbitrary languages using a pre-trained language model….” This is a **wrong** statement due to miscommunication. Our trained RED policy is able to handle any natural language commands.

In fact, we adopt an LSTM encoder to encode arbitrary natural language commands into a fixed-length sentence embedding. There are a total of 38,558 __different__ language commands in the training set. For human evaluation, the participants can send arbitrary language commands to cooperate with the policies. And we find that the RED policy can follow some unseen instructions — please refer to our updated website for examples and our code implementations. Meanwhile, a pre-trained language model can be potentially adopted as a more powerful language encoder than an LSTM, which we leave as future work.

Regarding the top 500 commands mentioned previously, this filtering process is only applied to Random and Human-Proxy command generators in our experiments. When testing the policy performances under random commands, we randomly sample commands from the top 500 commands every timestep. And the human proxy is trained to select a command from the top 500 commands based on the game observation. We have revised our paper to reflect these changes.

***

### New experiments.

We conduct a new experiment on updating the coefficient $\beta$ in Joint and RED methods, which leads to effective improvement. The results are listed in appendix D.2.

We also present two new evaluation metrics. The results are listed in appendix D.5

---

### Author Response · Authors · 2022-08-09
**We tackle natural language commands rather than a fixed command set, and have conducted additional experiments to analyze the influence of input words.**

We take reviewer WZeZ’s suggestion and analyze the influence of input words.  We evaluate how the agents perform on noisy commands, commands with Out-of-Vocabulary (OOV) words. We also present a clustering analysis of command representations. Please refer to ["Additional Results on noisy commands and the learned representation of the LSTM encoder"](https://openreview.net/forum?id=YYyAVk8TrOQ&noteId=8w3i84UC2lN) for more discussions. We will include these new results in the final draft.

---

### Meta-Review · Area_Chair_wP5X · 2022-08-25

**Recommendation:** Accept
**Confidence:** Certain

**Metareview:**

I thank the authors for their submission and active participation in the discussions. The paper investigates natural language constrained reward maximization. While this application paper is borderline, all reviewers unanimously agree that this paper's strengths outweigh its weaknesses. In particular, reviewers noted that the paper presents an interesting [8YRG,wnXo] and impactful [WZeZ] approach, evaluated using appropriate baselines [8YRG,wnXo], with insightful qualitative analyses [oT9Z], open-source code [oT9Z] and an overall clearly written paper [8YRG,9kih,WZeZ,wnXo]. Thus, I am recommending acceptance of the paper and encourage the authors to further improve their paper based on the reviewer feedback.

**Award:**

No

---

### Decision · Program_Chairs · 2022-09-14

Accept